# Bridging the Gap between Variational Inference and Stochastic Gradient MCMC in Function Space

**Mengjing Wu, Junyu Xuan, Jie Lu**
Australian Artificial Intelligence Institute, University of Technology Sydney, Australia
mengjing.wu@student.uts.edu.au; {junyu.xuan, jie.lu}@uts.edu.au

## Abstract

Traditional parameter-space posterior inference for Bayesian neural networks faces several challenges, such as the difficulty in specifying meaningful prior, the potential pathologies in deep models and the intractability for multi-modal posterior. To address these issues, functional variational inference (fVI) and functional Markov Chain Monte Carlo (fMCMC) are two recently emerged Bayesian inference schemes that perform posterior inference directly in function space by incorporating more informative functional priors. Similar to their parameter-space counterparts, fVI and fMCMC have their own strengths and weaknesses. For instance, fVI is computationally efficient but imposes strong distributional assumptions, while fMCMC is asymptotically exact but suffers from slow mixing in high dimensions. To inherit the complementary benefits of both schemes, this work proposes a novel hybrid inference method for the functional posterior inference. Specifically, it combines fVI and fMCMC successively by an elaborate linking mechanism to form an alternating approximation process. We also provide theoretical justification for the soundness of such a hybrid inference through the lens of Wasserstein gradient flows in the function space. We evaluate our method on several benchmark tasks and observe improvements in both predictive accuracy and uncertainty quantification compared to parameter/function-space VI and MCMC.

## 1 Introduction

Over the past few decades, Bayesian inference has emerged as a promising direction for deep learning (also known as Bayesian deep learning (Wilson & Izmailov, 2020)), offering improved predictive performance, principled estimation of epistemic uncertainty, and enhanced generalization (Neal, 1995; Gal et al., 2016; Wilson & Izmailov, 2020; Pielok et al., 2023). As one of the most representative models in Bayesian deep learning, Bayesian neural networks (BNNs) provide natural support for continual learning, reinforcement learning, and decision-making tasks. BNNs have found wide application in various safety-critical domains, including medical diagnosis such as Diabetic Retinopathy (Filos et al., 2019; Band et al., 2021), the dissolution prediction of planetary systems (Cranmer et al., 2021), and the classification of radio galaxies (Mohan & Scaife, 2024).

Due to the high-dimensional and multi-modal distributional properties, posterior inference for model parameters is remarkably challenging in modern BNNs (Izmailov et al., 2021). Typically, there are two popular classes of inference schemes: deterministic approximations, such as the variational inference (VI) (Blundell et al., 2015; Gal & Ghahramani, 2016), and sampling methods like Markov Chain Monte Carlo (MCMC) (Welling & Teh, 2011; Neal, 2012; Chen et al., 2014). However, the tricky prior issues in their parameter-space inference, including the non-interpretability and potential pathologies in deep models, e.g., the function samples of isotropy Gaussian prior over parameters tend to be horizontal as the depth of the network increases (Matthews et al., 2018; Tran et al., 2022), can severely impact the performance of BNNs in practical applications (Fortuin et al., 2022; Wild et al., 2022). These challenges have motivated researchers to explore Bayesian inference directly in function space. For example, based on the *Gaussian Processes* prior (Rasmussen & Williams, 2006; Titsias, 2009) which can effectively encode prior information about the periodicity, regularity, or smoothness through the corresponding kernel functions, there are several functional variational

inference (fVI) approaches (Sun et al., 2019; Ma & Hernández-Lobato, 2021; Rudner et al., 2022) have demonstrated improved prediction and uncertainty quantification compared to the parameter-space VI typically relies on non-informative isotropic Gaussian priors. In the realm of MCMC, Wu et al. (2024a) proposed a functional stochastic gradient MCMC (fMCMC) scheme, which lifts the parameter-space dynamics of BNNs onto function space, enabling the incorporation of more informative functional priors.

Despite their advantages, like their parameter-space counterparts, fVI and fMCMC have complementary strengths and limitations. fVI is fast and relatively easy to implement due to its explicit variational objective, but it is prone to bias and often relies on strong distributional assumptions about the structure and form of the variational posterior over parameters, such as using an oversimplified fully-factorized Gaussian for large models. Another drawback is that it tends to underestimate posterior variance (Zhang et al., 2018), leading to poor uncertainty estimation. On the other hand, fMCMC is non-parametric and can asymptotically generate exact samples from the true posterior measure, but it is computationally expensive and suffers from slow mixing in high-dimensional models. While stochastic gradient variants of MCMC are more scalable, they can introduce significant bias in posterior expectations (Izmailov et al., 2021).

Naturally, it is prospective to combine fVI and fMCMC to leverage their complementary strengths and further improve posterior inference in function space. There has been some work focused on how to connect VI and MCMC (Ahn et al., 2012; Domke, 2017; Habib & Barber, 2019). However, all these methods are performed in the parameter space, extending them to function spaces is not straightforward. For example, the structured MCMC proposed by Alexos et al. (2022) is based on an artificial partitioning structure of variational posterior over parameters, which is hardly applicable to infinite-dimensional posterior over functions. The tighter variational bound proposed by Salimans et al. (2015) in an expanded space is constructed by auxiliary random variables embedded from the intermediate MCMC transitions. However, extending this expanded variational posterior over parameters directly to an infinite-dimensional variational posterior over functions is not reasonable.

In this work, we propose a new hybrid inference method for posterior over functions that combines ideas from both fVI and fMCMC called *Functional Variational Inference MCMC (FVIMC)*. We bridge the theoretical gap between fVI and fMCMC and design a sophisticated linking mechanism that can alternate between variational optimization and sampling in a seamless manner. Specifically, we explore the relationship between these two schemes through the lens of Wasserstein gradient flows in the function space, revealing that they share the same probability evolution marginals in posterior inference. Thus, we innovatively alternate between fVI and fMCMC without changing the evolutionary path of the posterior, linking such two inference schemes together successively based on effective probability measure transformations to form a hybrid approximation process. This newly developed FVIMC incorporates meaningful functional priors and combines the strengths of both schemes: (1) the ability of fVI to quickly converge to local high-probability target regions and (2) the non-parametric nature of fMCMC, which allows it to explore multi-modal surfaces and prevent the posterior from collapsing to a local solution. Our main contributions are as follows:

- We develop a novel hybrid inference method that can efficiently link fVI and fMCMC successively to form an alternating approximation process, which can delicately inherit the benefits of both inference schemes and allow arbitrarily flexible and complex posterior over functions.

- We prove that the functional Langevin stochastic differential equation (SDE) behind the fMCMC and the probability flow ordinary differential equation (ODE) derived from the corresponding Wasserstein gradient flows of the fVI share the same probability evolution marginals, which can support the justification of our method.

- We perform a wide range of experiments to show the improved predictive performance and reliable uncertainty estimations of our method consistently compared to the competing VI and MCMC methods.

## 2 PRELIMINARIES

**Bayesian neural networks (BNNs)** Given a dataset $\mathcal{D} = \{x_i, y_i\}_{i=1}^N = \{\mathbf{X}_{\mathcal{D}}, \mathbf{Y}_{\mathcal{D}}\}$, where the training inputs $x_i \in \mathcal{X} \subseteq \mathbb{R}^p$ and the corresponding targets $y_i \in \mathcal{Y} \subseteq \mathbb{R}^c$, a Bayesian neural

network is a stochastic neural network characterized by random network parameters (weights and biases) denoted by a multivariate random variable $\mathbf{w} \in \mathbb{R}^k$ defined on a probability space $(\Omega, \mathcal{A}, P)$. Let $f(\cdot; \mathbf{w}) : \mathcal{X} \times \Omega \to \mathcal{Y}$ be the random function (product measurable) defined by a BNN on an infinite-dimensional function (Polish) space $\mathbb{H}$ with Borel $\sigma$-algebra $\mathcal{B}(\mathbb{H})$, which is $\mathcal{A}$ measurable for every $x \in \mathcal{X}$. The prior distribution for the network parameters $\mathbf{w}$ is denoted as $p_0(\mathbf{w})$. The likelihood is defined by a function $p$ as $\mathcal{Y} \times \mathbb{H} \to [0, \infty)$, $(\mathcal{Y}, f(\mathcal{X}; \mathbf{w})) \mapsto p(\mathcal{Y}|f(\mathcal{X}; \mathbf{w}))$, where $\mathcal{Y} \subseteq \mathbb{R}^c$ is Borel measurable. The likelihood evaluated on the training data is $p(\mathbf{Y}_\mathcal{D}|f(\mathbf{X}_\mathcal{D}; \mathbf{w}))$. The posterior distribution over parameters is then inferred with the unnormalised form as $p(\mathbf{w}|\mathcal{D}) \propto p(\mathbf{Y}_\mathcal{D}|f(\mathbf{X}_\mathcal{D}; \mathbf{w}))p_0(\mathbf{w})$. Given test data $x^*$, the predictive distribution for $y^*$ is obtained by integrating over the posterior as $p(y^*|x^*, \mathcal{D}) = \int p(y^*|f(x^*; \mathbf{w}))p(\mathbf{w}|\mathcal{D})\mathrm{d}\mathbf{w}$.

**Function-space VI** Suppose $\mathcal{P}(\mathbb{H})$ represents the space of Borel probability measures on $\mathcal{B}(\mathbb{H})$. To incorporate more meaningful prior information into posterior inference, fVI assigns a functional prior measure $P_0 \in \mathcal{P}(\mathbb{H})$, such as the classical Gaussian Processes (GP) prior denoted by $P_0 \sim \mathcal{GP}(\mathbf{m}, \mathbf{K})$ and perform variational inference in function spaces. The posterior measure $P_{f|\mathcal{D}} \in \mathcal{P}(\mathbb{H})$, induced by the functional prior and likelihood, is defined by the Radon–Nikodym derivative as $P_{f|\mathcal{D}}(\mathrm{d}f) \propto p(\mathbf{Y}_\mathcal{D}|f(\mathbf{X}_\mathcal{D}; \mathbf{w}))P_0(\mathrm{d}f)$ (Matthews et al., 2016; Lambley, 2023). The functional variational objective is to minimize the Kullback-Leibler (KL) divergence $\min_{Q_f} \mathrm{KL}[Q_f \| P_{f|\mathcal{D}}]$, where $Q_f \in \mathcal{P}(\mathbb{H})$ [1] is the approximate posterior measure induced by the approximate posterior distribution over parameters denoted by $q(\mathbf{w}; \boldsymbol{\theta})$, with variational parameters $\boldsymbol{\theta}$, and $q(\mathbf{w}; \boldsymbol{\theta})(\mathrm{d}\mathbf{w}) = Q_f(\mathrm{d}f)$. The pushforward of $q(\mathbf{w}; \boldsymbol{\theta})$, denoted by $\mathcal{T}_\# q(\mathbf{w}; \boldsymbol{\theta}) := Q_f$, where $\mathcal{T}_\#(\cdot)$ refers to the pushforward measure (Wild et al., 2022; Rudner et al., 2022). The functional evidence lower bound (ELBO) to be maximized in this framework is

$$
\begin{aligned}
\mathcal{L}_{Q_f} &:= \mathbb{E}_{Q_f}\left[\log p(\mathbf{Y}_\mathcal{D} \mid f(\mathbf{X}_\mathcal{D}; \mathbf{w}))\right] - \mathrm{KL}[Q_f \| P_0] \\
&= \mathbb{E}_{Q_f}\left[\log p(\mathbf{Y}_\mathcal{D} \mid f(\mathbf{X}_\mathcal{D}; \mathbf{w}))\right] - \sup_{\mathbf{X} \in \mathcal{X}_\mathbb{N}} \mathrm{KL}[Q_f(f^\mathbf{X}) \| P_0(f^\mathbf{X})],
\end{aligned} \tag{1}
$$

where $\mathcal{X}_\mathbb{N} \doteq \bigcup_{n \in \mathbb{N}} \{\mathbf{X} \in \mathcal{X}_n \mid \mathcal{X}_n \subseteq \mathbb{R}^{n \times p}\}$ is the set of all finite marginal measurement points in the input domain, $f^\mathbf{X}$ is the function value evaluated at $\mathbf{X}$ (Sun et al., 2019).

**Function-space MCMC** Wu et al. (2024a) explored the dynamics of BNNs in function space and proposed a novel fMCMC scheme. This functional MCMC scheme is based on the potential energy functional designed in terms of the posterior over functions $P_{f|\mathcal{D}}$, which effectively incorporates functional priors while ensuring that the target posterior over functions remains the stationary distribution. The specific functional Langevin dynamics for $f(\cdot; \mathbf{w})$ on $\mathbb{H}$ is given by:

$$
\begin{aligned}
\mathrm{d}f_t(\cdot; \mathbf{w}) &= \mu(f_t(\cdot; \mathbf{w}))\mathrm{d}t + \sigma(f_t(\cdot; \mathbf{w}))\mathrm{d}B_t \\
&= \left[-(\nabla_\mathbf{w} f_t)^T(\nabla_\mathbf{w} f_t)\nabla_f\left(-\log p(\mathbf{Y}_\mathcal{D}|f_t(\mathbf{X}_\mathcal{D}; \mathbf{w})) + I_0(f_t)\right) + H_\mathbf{w} f_t\right]\mathrm{d}t + \sqrt{2}(\nabla_\mathbf{w} f_t)^T \mathrm{d}B_t,
\end{aligned} \tag{2}
$$

where $I_0(f)$ denotes the *Onsager–Machlup* (OM) functional for $P_0$ ( heuristically interpreted as the negative logarithm of $P_0$ (Lambley, 2023)), $B$ is a Wiener process (Brownian motion), $\nabla_\mathbf{w} f$ is the Fréchet derivative of $f(\cdot; \mathbf{w})$ w.r.t. network parameters $\mathbf{w}$, and $H_\mathbf{w} f$ denotes the second-order Fréchet derivative. Additionally, leveraging the Itô Lemma (see in Appendix A), they derived the corresponding diffusion process for the network parameters. The specific discretization update rule for sampling the network parameters under this functional stochastic gradient Langevin dynamics (fSGLD) is

$$
\mathbf{w}_{t+1} = \mathbf{w}_t - \epsilon_t \nabla_\mathbf{w} f\left[-\frac{N}{n}\sum_{i=1}^n \nabla_f \log p(\mathbf{y}_i|f_t(\mathbf{x}_i; \mathbf{w}_t)) - \nabla_f \mathbf{x}_\mathcal{M} \log P_0(f_t^{\mathbf{X}_\mathcal{M}})\right] + \sqrt{2\epsilon_t}\eta_t, \tag{3}
$$

where $\epsilon_t$ is the decreasing step-size, $\eta_t$ is a standard Gaussian noise, $n$ is the mini-batch size, $\mathbf{X}_\mathcal{M} \overset{\text{def}}{=} [\mathbf{x}_1, \ldots, \mathbf{x}_M]^T$ are the finite measurement points, and $f^{\mathbf{X}_\mathcal{M}}$ are the corresponding function values evaluated at $\mathbf{X}_\mathcal{M}$.

---

[1]Note that without loss of generality, we assume that $Q_f \in \mathcal{P}(\mathbb{H})$ is dominated by $P_0$ (therefore also dominated by $P_{f|\mathcal{D}}$) to avoid technical difficulties in the definition.

## 3   OUR METHOD

In this section, we propose to bridge the theoretical gap between these two functional inference schemes and develop a novel hybrid inference method for posterior measure over functions. We start from an initial random function mapping defined by a BNN denoted by $f_{s0}(\cdot; \mathbf{w}_{s0})$, given a simple initial approximate posterior over parameters $q_{s0}(\mathbf{w}_{s0}; \boldsymbol{\theta}_{s0})$, e.g., a factorized Gaussian has $\boldsymbol{\theta}_{s0} = \{\mu_{s0}, \boldsymbol{\Sigma}_{s0}\}$, which induces the posterior measure over functions denoted by $Q_{f_{s0}}$. Note that this factorized Gaussian will be modified later rather than being fixed, as in fVI, so it does not impose a strict restriction. Our functional hybrid inference method is formed by the following alternating approximation stages:

- **V-Stage:** The first stage starts by executing the fVI through maximizing the functional ELBO in Equation (1) to obtain the updated approximate posterior measure as $Q_{f_{s1}}$, and the corresponding variational posterior distribution over parameters is $q_{s1}(\mathbf{w}_{s1}; \boldsymbol{\theta}_{s1})$ with $\boldsymbol{\theta}_{s1} = \{\mu_{s1}, \boldsymbol{\Sigma}_{s1}\}$.

- **M-Stage:** In this stage, starting from the randomly sampled initial point $\mathbf{w}_{s1} \sim \mathcal{N}(\mu_{s1}, \boldsymbol{\Sigma}_{s1})$ from the last variational stage, we perform stochastic gradient fMCMC using fSGLD in Equation (3). Let $\mathbf{w}_{s2_{\{1:N\}}}$ denotes the collected $N$ samples of network parameters, the corresponding function samples is as $f_{s2_{\{1:N\}}}$. The distribution over parameters for the collected network parameters samples is denoted by $q_{s2}(\mathbf{w}_{s2})$, which is distributional assumption free and implicit, and the corresponding probability measure over functions is as $Q_{f_{s2}}$.

Our idea is to alternate between the two stages, but one challenge arises: $q_{s2}(\mathbf{w}_{s2})$ from the M-Stage is non-parametric and lacks the explicit distribution form required by the V-Stage. To address this, we reconsider the relationship between $Q_{f_{s2}}$ and $Q_{f_{s1}}$, where the latter comes from the last V-Stage with explicit parametric $q_{s1}(\mathbf{w}_{s1}; \boldsymbol{\theta}_{s1})$. We can simply consider $Q_{f_{s1}}$ as the input to the M-Stage and $Q_{f_{s2}}$ as the output. Thus, $Q_{f_{s2}}$ is a refined posterior compared to $Q_{f_{s1}}$. Given this relationship, we propose treating $Q_{f_{s2}}$ as a transformation of $Q_{f_{s1}}$, where the transformation follows a nonlinear, parameterized form. By uncovering this transformation, we can derive an explicit parametric $q_{s2}(\mathbf{w}_{s2})$ with the aid of $q_{s1}(\mathbf{w}_{s1}; \boldsymbol{\theta}_{s1})$. We define the following stage for this transformation learning:

- **T-Stage:** Let $F_{\lambda_1}^{-1} : f_{s2}(\cdot; \mathbf{w}_{s2}) \rightarrow f_{s1}(\cdot; \mathbf{w}_{s1})$ be an invertible bijective function parametrized by $\lambda_1$ (with inverse $F_{\lambda_1}$), where $f_{s2}(\cdot; \mathbf{w}_{s2})$ and $f_{s1}(\cdot; \mathbf{w}_{s1})$ denote the obtained random function mappings from the M-Stage and V-Stage, respectively. According to the change of variable formula, we have

$$Q_{f_{s2}}(f_{s2}^{\mathbf{X}_{\mathcal{M}}}) = Q_{f_{s1}}(F_{\lambda_1}^{-1}(f_{s2}^{\mathbf{X}_{\mathcal{M}}})) \left| \det \frac{\partial F_{\lambda_1}}{\partial F_{\lambda_1}^{-1}(f_{s2}^{\mathbf{X}_{\mathcal{M}}})} \right|^{-1} = Q_{f_{s1}}(F_{\lambda_1}^{-1}(f_{s2}^{\mathbf{X}_{\mathcal{M}}})) \left| \det \frac{\partial F_{\lambda_1}^{-1}}{\partial f_{s2}^{\mathbf{X}_{\mathcal{M}}}} \right|$$

(4)

and

$$\log Q_{f_{s2}}(f_{s2}^{\mathbf{X}_{\mathcal{M}}}) = \log Q_{f_{s1}}(F_{\lambda_1}^{-1}(f_{s2}^{\mathbf{X}_{\mathcal{M}}})) + \log \left| \det \frac{\partial F_{\lambda_1}^{-1}}{\partial f_{s2}^{\mathbf{X}_{\mathcal{M}}}} \right|,$$

(5)

where $\mathbf{X}_{\mathcal{M}} \stackrel{\text{def}}{=} [\mathbf{x}_1, \ldots, \mathbf{x}_M]^{\mathrm{T}}$ are finite randomly sampled measurement points and $f_{s2}^{\mathbf{X}_{\mathcal{M}}}$ are the function evaluations at $\mathbf{X}_{\mathcal{M}}$. Based on the local linearization (Rudner et al., 2022), the implicit functional $Q_{f_{s1}}$ is approximated as a Gaussian process given by $\mathcal{GP}(f_{s1}|f_{s1}(\cdot, \mu_{s1}), \mathcal{J}_{\mu_{s1}} \boldsymbol{\Sigma}_{s1} \mathcal{J}_{\mu_{s1}}^{T})$, where $\mathcal{J}_{\mu_{s1}}$ denotes the Jacobian $\frac{\partial f_{s1}(\cdot, \mathbf{w}_{s1})}{\partial \mathbf{w}_{s1}}|_{\mathbf{w}_{s1} = \mu_{s1}}$, and the finite marginal distribution of $Q_{f_{s1}}$ is reduced to an analytical Gaussian distribution. Given the function samples $f_{s2_{\{1:N\}}}^{\mathbf{X}_{\mathcal{M}}}$ on $\mathbf{X}_{\mathcal{M}}$ from M-Stage, we can obtain the optimal bijective function $F_{\lambda_1^*}^{-1}$ by maximizing Equation (5) and transform $f_{s2}(\cdot; \mathbf{w}_{s2})$. With $F_{\lambda_1^*}^{-1}$ in hand, we can obtain the explicit form of $Q_{f_{s2}}$ as $F_{\lambda_1^*} \circ Q_{f_{s1}}$, where $F_{\lambda_1^*} \circ Q_{f_{s1}}$ denotes the pushforward relationship as shown in Equation (4).

With the help of this T-Stage, we can successfully link the above V-Stage and M-Stage and form a chain as follows:

$$\text{V-Stage} \xrightarrow{Q_{f_{s1}}} \text{M-Stage} \xrightarrow{f^{\mathbf{x}_{\mathcal{M}}}_{s2\{1:N\}}} \text{T-Stage} \xrightarrow{Q_{f_{s2}}} \text{V-Stage} \xrightarrow{F_{\lambda_1^*} \circ Q_{f_{s3}}} \cdots \longrightarrow \text{V-Stage} \,. \quad (6)$$

The pseudocode for the above procedure (named FVIMC) is shown in Algorithm 1 in Appendix C. Let us have an in-depth analysis of this procedure. Because each T-Stage in this hybrid inference process yields a new explicit form for the approximate posterior obtained from the M-Stage based on the approximate posterior from the previous V-Stage, and this will be used for the following V-Stage, the form of the final posterior from the above procedure would no longer be a (initialized) factorized Gaussian, and we have the following result.

**Proposition 3.1.** *The final approximate posterior measure is in a composited form as $F_{\lambda_{(K-1)/2}^*} \circ \cdots F_{\lambda_2^*} \circ F_{\lambda_1^*} \circ Q_{f_{sK}}$ with a total of $K$ alternating approximate V-Stages and M-Stages, which is arbitrarily complex and preferably flexible.*

The derivation is given in Appendix B.1. This proposition demonstrates that, although we initially set a simple approximate posterior, the final result can still be flexible enough to match a complex posterior. This validates our ability to overcome the strong distributional assumption of the variational posterior in fVI. The key to achieving this lies in the M-Stage, which introduces non-parametric updates to the posterior. By alternating between the V-Stage and M-stage with the support of the T-Stage, the process continuously refines the posterior approximation. Specifically, the M-Stage introduces samples that capture the multi-modal nature of the true posterior, while the V-Stage applies optimization to tighten the approximation. Through this alternating and successive process, the approximate posterior evolves to become more expressive and accurate, leveraging the strengths of both schemes—fVI's efficiency in local regions and fMCMC's ability to explore complex distributions.

Another important question arises: can we safely link the M-Stage and V-Stage? Specifically, how can we ensure consistent convergence paths if these two stages evolve in different directions and ultimately converge to distinct optimums? To address this concern, we present the following result:

**Proposition 3.2.** *The functional Langevin SDE defined in Equation* (2) *and the probability flow ODE derived from the Wasserstein gradient of the functional variational inference in Equation* (1) *share the same probability measure evolution marginals if they evolve from the same initial point.*

The proof is given in Appendix B.2. This proposition guarantees the soundness of our proposed hybrid FVIMC since the approximate posterior measures of fSGLD and fVI share the same evolution marginals. Therefore, it is reasonable to link these two schemes successively to form an alternating approximation process.

**The form of the invertible transformations** The parametric form of transformation $F_\lambda(\cdot)$ in the T-Stage plays a crucial role in determining the approximation accuracy to the samples from the M-Stage and, consequently, the flexibility of the posterior modelling capability. In general, increasing complexity improves approximation. However, more complex transformations also increase computational difficulty, particularly when calculating the determinant of the Jacobian during optimization. Therefore, designing $F_\lambda(\cdot)$ requires a balance between modelling flexibility and computational feasibility. Here, we use three classes of flexible and tractable invertible bijections: linear, non-linear Tanh and Sigmoid, respectively:

$$y = \frac{1}{a} f(x; \mathbf{w}) - \frac{1}{a} b(x; \omega), \quad (7)$$

$$y = a_2 \operatorname{Tanh}\left(a_1 f(x; \mathbf{w}) + b_1(x; \omega_1)\right) + b_2(x; \omega_2), \quad (8)$$

$$y = a_2 \operatorname{Sigmoid}\left(a_1 f(x; \mathbf{w})\right) + b(x; \omega), \quad (9)$$

where $f(x; \mathbf{w})$ denotes function mappings, $a$ is a scalar parameter and $b(x; \omega)$ denotes a bias function. The computation of the Jacobian for all three transformations is straightforward and does not involve the Jacobian of function $b(x; \omega)$, so it can be arbitrarily complex and we make them deep neural networks with parameters $\omega$ to achieve high expression ability. Then the parameters of the bijections are denoted as $\lambda = \{a, \omega\}$. The specific form for each transformation need not be fixed.

For flexibility, we can implement an automatic selection mechanism that chooses the form with the highest likelihood value during the optimization. This allows the model to adaptively select the most suitable transformation for different function mappings. More details about the Jacobians and inverse bijections are shown in Appendix D.

**Avoid the potential problematic KL divergence for fVI**  The KL divergence in fVI is found to be potentially problematic (Burt et al., 2020) (see Appendix A). Hence, following the latest idea in the area (Knoblauch et al., 2022), we propose to use the Wasserstein distance to replace the KL divergence in the V-Stage of our FVIMC with the alternative functional variational objective as $\mathcal{L}_W := -\mathbb{E}_{Q_f}\left[\log p(\mathbf{Y}_\mathcal{D} \mid f(\mathbf{X}_\mathcal{D}; \mathbf{w}))\right] + W_1(Q_f, P_0) + M_2(Q_f, P_0)$, where $W_1(Q_f, P_0)$ is the 1-Wasserstein distance between $Q_f$ and $P_0$ that admits the dual form definition as $\sup_{\|\phi\| \leq 1} \mathbb{E}_{\mathbf{x} \sim Q_f} \phi(\mathbf{x}) - \mathbb{E}_{\mathbf{y} \sim P_0} \phi(\mathbf{y})$, and $M_2(Q_f, P_0) = |var(Q_f) - var(P_0)|$ is an enhanced second-order moment matching term to preserve the uncertainty information encoded in the functional prior, which is found to be weak in the naive 1-Wasserstein distance (Wu et al., 2024b). Compared to the KL divergence-based fVI, which is severely limited by the requirement for explicitly accessible densities, the Wasserstein distance-based functional variational objective offers significantly more flexibility. This is because it relies entirely on a sampling-based procedure, eliminating the need for the closed-form of $Q_f$. At the same time, Proposition 3.2 still holds under this replacement in certain cases (see Appendix G.5 for more discussions and empirical evidence).

## 4  RELATED WORK

**Functional posterior inference in BNNs**  Due to unresolved deficiencies of parameter-space inference for BNNs, recent work has shifted toward performing Bayesian inference directly in function space. For functional variational inference, Sun et al. (2019) introduced a functional ELBO that leverages more informative GP priors and explicitly minimizes the KL divergence between the variational posterior and the true posterior in function space. They proved that the KL divergence between two infinite-dimensional stochastic processes corresponds to the supremum of the KL divergence over all finite marginal distributions. However, employing KL divergence in function spaces requires careful consideration as it presents several definitional and estimation challenges (Burt et al., 2020). For example, the variational posterior measure must be dominated by the functional prior to ensure the existence of the Radon-Nikodym derivative, which defines the posterior measure (Matthews et al., 2016; Wild et al., 2022). If this condition is unmet, this functional KL divergence may become ill-defined, e.g., the KL divergence between two BNNs formed by different network architectures can be infinite (Ma & Hernández-Lobato, 2021). Additionally, Rudner et al. (2022) pointed out the analytical intractability of the supremum over marginal KL divergences and proposed a more practical, well-defined functional variational objective. Their method is based on approximating the posterior and prior over functions as Gaussian processes via local linearization. Considering the limitations of functional KL divergence, Tran et al. (2022) proposed matching a BNN prior with an interpretable GP prior using the 1-Wasserstein distance (Kantorovich, 1960), while Wild et al. (2022) introduced a generalized functional variational objective for Gaussian measures based on the 2-Wasserstein distance. For functional MCMC, Wu et al. (2024a) investigated the diffusion process of BNNs in function space and designed a new functional stochastic gradient MCMC scheme, which can incorporate functional priors and guarantee that the stationary distribution of the resulting functional dynamics aligns with the target posterior distribution over functions.

**Connections between VI and MCMC in parameter space**  There has also been some work aimed at hybridizing VI and MCMC in parameter space. Most of these efforts focus on sampling from an approximate posterior, rather than the true posterior, typically on a tempered or structured evidence lower bound. Alternatively, they optimize variational parameters using auxiliary variables derived from embedded MCMC schemes to achieve a tighter variational bound. For example, Ahn et al. (2012) proposed a Stochastic Gradient Fisher Scoring (SGFS) algorithm that uses the inverse Fisher information matrix to sample from a Gaussian approximation of the posterior based on stochastic gradient Langevin dynamics. Salimans et al. (2015) integrated Markov transition chains into variational lower bound as auxiliary variables to improve posterior approximation. Domke (2017) derived two bounds for the variational divergence and introduced a hybrid algorithm based on it that directly interpolates between variational inference and Langevin dynamics. Hoffman & Ma (2020) find that Langevin dynamics can implicitly track the nonparametric normalizing flow of variational

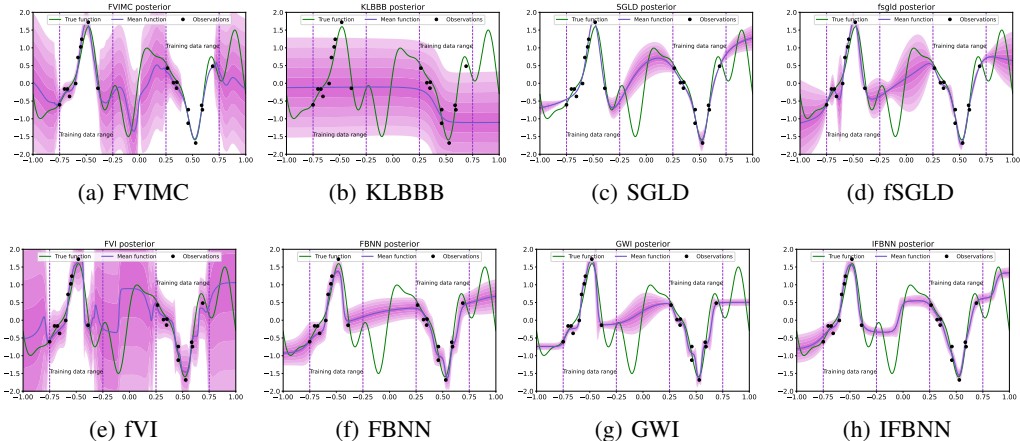

Figure 1: Learning polynomial curves. The green line is the ground true function and the blue lines correspond to mean approximate posterior predictions. Black dots denote 20 training points; shadow areas represent the predictive standard deviations. The top row is our FVIMC, parameter-space VI/SGLD and functional fSGLD, and the bottom row shows the results of four function-space VI baselines. For more experimental details, see Appendix E. The complete inference process of FVIMC is shown in Appendix F.

inference, reinterpreting black-box variational inference as a parametric approximation to Langevin dynamics. Moreover, Liu et al. (2019) developed a framework on the Wasserstein space, casting general MCMC dynamics on it as the fiber-gradient Hamiltonian flow. Alexos et al. (2022) proposed a structured MCMC scheme that samples from an approximate posterior under distributional factorization constraints. However, all of these methods are restricted to parameter space, making it challenging to extend them directly to function space.

## 5 EXPERIMENTS

We evaluate FVIMC on a variety of benchmark tasks, including a synthetic extrapolation, multivariate regressions on UCI datasets, contextual bandits and image classification tasks. We compare the performance of FVIMC with several competing parameter/function-space VI and MCMC baselines, including the benchmark parameter-space VI method BBB (Blundell et al., 2015) denoted by KLBBB, and four competing function-space VI methods: FBNN (KL divergence-based fVI) (Sun et al., 2019), GWI(Wild et al., 2022), IFBNN(Wu et al., 2023) and our proposed Wasserstein distance-based fVI denoted as fVI. Also, for MCMC methods, we compare with the classic parameter-space stochastic gradient Langevin dynamics denoted by SGLD (Welling & Teh, 2011) and the functional fSGLD (Wu et al., 2024a). We obtain the improved predictive performance and uncertainty quantification, indicating that FVIMC can efficiently incorporate meaningful knowledge from the functional prior and inherit the strengths of both fVI and fSGLD schemes.

### 5.1 EXTRAPOLATION ON SYNTHETIC DATA

We begin by verifying the data fitting and uncertainty quantification capabilities of our method through a 1-D oscillation curve extrapolation experiment using a synthetic dataset. Consider a polynomial target function: $y = \sin(3\pi x) + 0.3\cos(9\pi x) + 0.5\sin(7\pi x) + \epsilon$ with noise $\epsilon \sim \mathcal{N}(0, 0.5^2)$, we randomly sampled 10 points from Uniform$(-0.75, -0.25)$ and another 10 points from Uniform$(0.25, 0.75)$ as input data. The model used is a fully-connected neural network with two hidden layers. We apply the same GP prior with the RBF kernel, pre-trained on the input dataset, for all functional methods. For all parameter-space methods that cannot specify a meaningful prior, we place the default isotropic Gaussian prior over network parameters. For our FVIMC, we employ a hybrid inference process consisting of 21 alternating V-Stages and M-Stages, along with 10 intermediate T-Stages. Specifically, this process includes 11 V-Stages and 10 M-Stages,

Table 1: The table shows the results of average RMSE for multivariate regression on UCI datasets. We split each dataset randomly into 90% training data and 10% test data, and this process is repeated 10 times to ensure validity. Bold indicates statistically significant best results ($p < 0.01$ with t-test).

| | RMSE | | | | | | |
|---|---|---|---|---|---|---|---|
| | Yacht | Boston | Concrete | Energy | Wine | Kin8nm | Protein |
| FVIMC | **0.37 ± 0.13** | **0.36 ± 0.07** | **0.40 ± 0.04** | **0.23 ± 0.02** | **0.68 ± 0.05** | **0.57 ± 0.02** | 0.95 ± 0.03 |
| fVI | 0.45 ± 0.10 | 0.40 ± 0.08 | 0.44 ± 0.04 | 0.27 ± 0.02 | 0.69 ± 0.05 | 0.61 ± 0.02 | 1.01 ± 0.02 |
| GWI | 2.20 ± 0.08 | 1.74 ± 0.05 | 1.30 ± 0.05 | 1.46 ± 0.04 | 1.68 ± 0.06 | 1.19 ± 0.02 | 1.33 ± 0.01 |
| FBNN | 1.52 ± 0.08 | 1.68 ± 0.12 | 1.27 ± 0.05 | 1.35 ± 0.06 | 1.53 ± 0.05 | 1.45 ± 0.07 | 1.50 ± 0.03 |
| IFBNN | 1.24 ± 0.10 | 1.44 ± 0.09 | 1.07 ± 0.07 | 1.19 ± 0.05 | 1.21 ± 0.05 | 1.12 ± 0.02 | 1.16 ± 0.01 |
| KLBBB | 2.13 ± 0.09 | 1.92 ± 0.07 | 1.78 ± 0.06 | 1.78 ± 0.07 | 1.86 ± 0.07 | 1.79 ± 0.03 | 1.80 ± 0.01 |
| SGLD | 1.09 ± 0.10 | 1.23 ± 0.06 | 1.10 ± 0.07 | 1.04 ± 0.06 | 1.08 ± 0.09 | 1.20 ± 0.02 | 1.12 ± 0.01 |
| fSGLD | 0.41 ± 0.10 | 0.36 ± 0.09 | 0.45 ± 0.04 | 0.24 ± 0.03 | 0.71 ± 0.05 | 0.74 ± 0.02 | **0.84 ± 0.01** |

with each V-Stage trained for 1000 epochs and each M-Stage run for 500 iterations. We use two different bijections for the transformations of probability measures in the T-Stage after each M-Stage: the linear transformation in Equation (7) and the non-linear Tanh bijection in Equation (8). The bias function for both bijections is parametrized by a neural network with two hidden layers. We implement the likelihood-based auto-selection mechanism to choose between these two invertible transformations. For fair comparisons, we run 16000 epochs for all other parameter/function-space VI methods. For all sampling methods, we use 2000 burn-in iterations and 14000 iterations for 140 samples (to reduce correlations between samples, we draw separated samples every 100 epochs). The results are shown in Figure 1. The parameter-space VI method KLBBB in Figure 1(b) fails to fit the target function. The two MCMC methods could recover the key polynomial trend of the curve in the observation range, while the uncertainty quantization in the unseen areas is not justified. Similarly, the three function-space VI methods in Figure 1(f), Figure 1(g), and Figure 1(h) also severely underestimates the predictive uncertainty, which is a typical downside of VI (Zhang et al., 2018; Alexos et al., 2022). In contrast, our FVIMC and the Wasserstein distance-based fVI show better predictive performances and improved uncertainty estimations. In particular, the hybrid FVIMC in Figure 1(a) exhibits stronger ability than the pure variational fVI in Figure 1(e), e.g., the smoother fitting curves, and can capture critical features of the non-observed middle range $[-0.25, 0.25]$, which demonstrates the powerful advantages of combining fVI and fMCMC for a hybrid inference. See Appendix H for more detailed ablation studies about the impact of kernel functions in functional GP priors and the form of the bijections for probability transformations.

## 5.2 UCI REGRESSION

In this experiment, we evaluate the predictive performance of FVIMC for multivariate regression tasks on 7 real-world UCI datasets: *Yacht, Boston, Concrete, Energy, Wine, Kin8nm and Protein*. We use a 2- hidden-layer fully connected neural network. For FVIMC, we employ a total of five alternating V-Stages and M-Stages to form the hybrid approximation process. Each V-Stage and M-Stage is trained for 400 epochs, resulting in a total of 2000 iterations. Therefore, we run 2000 epochs for all other VI methods for fair comparison. For the two sampling methods, we run 500 iterations for the burn-in stage and collect 15 samples in the following 1500 iterations. We report the average root mean square error (RMSE) results in Table 1. Our FVIMC achieves better accuracy than all other VI and MCMC baselines in 6/7 datasets, which illustrates the impressive predictive power of our hybrid method. For NLL results shown in Table 2, FVIMC outperforms other methods on 5/7 datasets, demonstrating its competitive uncertainty quantification ability. Moreover, the analyses of the mixing time, computational complexity and transformation errors in the T-Stage are presented in Appendix G, where our FVIMC exhibits high convergence speed and training stability.

## 5.3 CONTEXTUAL BANDITS

In this section, we examine the ability to guide exploration-exploitation in contextual bandits, where uncertainty modelling is crucial in sequential decision-making for such downstream tasks. In this problem, the agent interacts with an unknown environment repeatedly and chooses an optimal action to maximize the reward given the context in each round of interaction. Thompson sampling is a classic technique applicable to this scenario Thompson (1933). It samples a model configuration from

Table 2: The table shows the results of average NLL for multivariate regression on UCI datasets. We split each dataset randomly into 90% training data and 10% test data, and this process is repeated 10 times to ensure validity. Bold indicates statistically significant best results ($p < 0.01$ with t-test).

| | NLL | | | | | | |
| | YACHT | BOSTON | CONCRETE | ENERGY | WINE | KIN8NM | PROTEIN |
|---|---|---|---|---|---|---|---|
| FVIMC | **-5.03 ± 1.13** | **-3.74 ± 0.48** | **-5.00 ± 0.34** | **-5.70 ± 0.39** | **-3.77 ± 0.40** | -0.31 ± 0.65 | -2.07 ± 0.36 |
| FVI | -3.07 ± 0.66 | -2.66 ± 0.27 | -3.03 ± 0.22 | -4.20 ± 0.25 | -2.25 ± 0.20 | -0.59 ± 0.54 | **-2.30 ± 0.30** |
| GWI | 0.11 ± 0.76 | -1.04 ± 0.68 | -0.68 ± 0.49 | -2.03 ± 0.39 | 0.70 ± 0.16 | **-2.60 ± 0.24** | -1.58 ± 0.23 |
| FBNN | -0.77 ± 0.86 | -1.19 ± 0.76 | -1.00 ± 0.52 | -2.14 ± 0.47 | 0.52 ± 0.14 | -2.45 ± 0.62 | -1.49 ± 0.24 |
| IFBNN | -1.25 ± 1.21 | 0.32 ± 0.30 | -0.39 ± 0.33 | -1.78 ± 0.36 | 0.26 ± 0.15 | -1.01 ± 0.14 | -2.13 ± 0.34 |
| KLBBB | 2.51 ± 0.16 | 2.07 ± 0.12 | 2.61 ± 0.17 | 2.17 ± 0.14 | 2.15 ± 0.13 | 2.61 ± 0.07 | 2.22 ± 0.02 |
| SGLD | -0.37 ± 0.08 | -0.60 ± 0.11 | -0.54 ± 0.05 | -0.58 ± 0.09 | -0.90 ± 0.12 | -0.60 ± 0.04 | -0.60 ± 0.01 |
| FSGLD | -2.46 ± 0.28 | -2.20 ± 0.20 | -1.63 ± 0.14 | -2.25 ± 0.25 | -2.53 ± 0.15 | -2.24 ± 0.09 | -2.10 ± 0.08 |

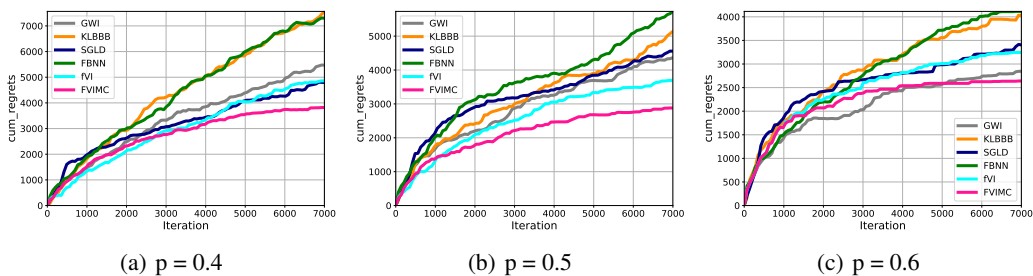

(a) p = 0.4      (b) p = 0.5      (c) p = 0.6

Figure 2: Comparisons of cumulative regrets of FVIMC, fVI, FBNN, GWI, KLBBB, SGLD for contextual bandit task on the Mushroom dataset. Lower represents better performance.

the current posterior first and adaptively chooses an optimal action under this sampled configuration for the current context, then updates the posterior based on the corresponding observed reward.

We compare FVIMC with several baselines on the UCI Mushroom dataset with 8124 instances. In each instance, the mushroom is labelled as edible or poisonous and has 22 features as the input context. The agent can observe mushroom features in each interaction and choose to eat or reject a mushroom. We follow the basic settings by (Blundell et al., 2015) and consider three different reward patterns: if the agent eats an edible mushroom, it receives a reward of 5; otherwise, a reward of 0 if the agent rejects the edible mushroom. For the poisonous situation, if the agent chooses to eat, it receives a reward of -35 with probabilities of 0.4, 0.5, and 0.6, respectively, to form three different patterns; otherwise, a reward of 0 for rejection. Suppose an oracle always eats an edible mushroom and rejects the poisonous ones. The cumulative regrets with respect to the reward achieved by the oracle can measure the exploration-exploitation ability of an agent. For FVIMC, we run five alternating V-Stages and M-Stages with a total of 7000 iterations. And we run 7000 epochs for all other baselines with batch size 64. The cumulative regrets for three reward patterns are shown in Figure 2, where our FVIMC consistently achieves the lowest regrets, which indicates its reliable uncertainty quantification in decision-making.

## 5.4 IMAGINE CLASSIFICATION AND OOD DETECTION

We demonstrate the scalability of FVIMC on high-dimensional image classification tasks. We test the in-distribution predictive performance and out-of-distribution (OOD) detection ability on MNIST (LeCun et al., 2010) and FashionMNIST (Xiao et al., 2017). For our FVIMC, we use five alternating V-Stages and M-Stages with a total of 320 iterations. Therefore, we run 320 epochs for all other VI methods. For sampling methods, we run for 300 burn-in iterations and then collect 20 samples. We report the test classification errors (%) for predictive performance and the area under the curve (AUC) of OOD detection pairs FashionMNIST/MNIST, MNIST/FashionMNIST based on predictive entropies in Table 3. Our FVIMC outperforms all parameter/function-space VI and MCMC baselines for classification accuracy and demonstrates competitive OOD detection ability.

Table 3: Image classification and OOD detection performance.

| | MNIST | | FMNIST | |
|---|---|---|---|---|
| MODEL | TEST ERROR | AUC | TEST ERROR | AUC |
| FVIMC | **4.95 ± 0.00** | **0.832 ± 0.00** | **14.57 ± 0.00** | 0.800 ± 0.00 |
| FVI | 5.49 ± 0.00 | 0.815 ± 0.01 | 14.99 ± 0.00 | 0.831 ± 0.05 |
| GWI | 6.23 ± 0.00 | 0.826 ± 0.03 | 15.24 ± 0.00 | 0.459 ± 0.07 |
| FBNN | 5.06 ± 0.00 | 0.787 ± 0.05 | 15.44 ± 0.00 | 0.809 ± 0.03 |
| IFBNN | 5.00 ± 0.00 | 0.703 ± 0.30 | 15.64 ± 0.00 | **0.833 ± 0.03** |
| KLBBB | 5.30 ± 0.00 | 0.827 ± 0.03 | 15.26 ± 0.00 | 0.782 ± 0.00 |
| FSGLD | 5.21 ± 0.00 | 0.756 ± 0.01 | 15.99 ± 0.00 | 0.810 ± 0.00 |

## 6 CONCLUSION

We propose a hybrid inference method for BNNs in function space, combining the strengths of both functional variational inference (fVI) and functional MCMC (fMCMC) schemes. To achieve this, we develop a hybrid approximation process that alternately and successively links these two schemes, allowing the process to inherit the advantages of both while constructing arbitrarily flexible and complex approximate posteriors. We prove that the functional Langevin SDE and the probability flow ODE derived from the Wasserstein gradient flows of the functional variational inference share the same probability evolution marginals. This ensures the theoretical soundness of the hybrid approach. Empirically, we show the improved predictive performance and uncertainty estimation of our method on a range of tasks.

## ACKNOWLEDGMENTS

This work is supported by the Australian Research Council under the Discovery Early Career Researcher Award DE200100245.

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

## A    FURTHER BACKGROUND

**Itô Lemma**    Itô Lemma (Itô, 1951) is a fundamental result in stochastic calculus to find the differential of a function of a stochastic process, which serves as the stochastic calculus counterpart of the chain rule. For an Itô diffusion, let $f(X_t)$ be an arbitrary twice differentiable scalar function of real variables $X_t$, then the differential of $f(X_t)$ can be derived from the Taylor series expansion of the function as

$$\mathrm{d}f(X_t) = \left( \mu(X_t)\frac{\partial f}{\partial x} + \frac{\sigma^2(X_t)}{2}\frac{\partial^2 f}{\partial x^2} \right)\mathrm{d}t + \sigma(X_t)\frac{\partial f}{\partial x}\mathrm{d}B_t, \tag{10}$$

which implies that $f(X_t)$ is itself an Itô diffusion (Brzeźniak et al., 2008).

**Change of variable formula and normalizing flows**    A normalizing flow is a powerful transformation that can generate highly flexible and complex probability density from a simple initial one through a sequence of bijective invertible mappings (Tabak & Turner, 2013; Kobyzev et al., 2020; Papamakarios et al., 2021). Let $\xi \in \mathbb{R}^d$ be a random variable with probability density $p_\xi(\xi)$, and $F : \mathbb{R}^d \to \mathbb{R}^d$ is an invertible mapping (with inverse $F^{-1}$). Then, the density of the transformed variable $\eta = F(\xi)$ can be obtained using the change of variable formula (Dinh et al., 2017) as

$$p_\eta(\eta) = p_\xi(\xi)\left| \det \frac{\partial F^{-1}}{\partial \eta} \right| = p_\xi(\xi)\left| \det \frac{\partial F}{\partial \xi} \right|^{-1}. \tag{11}$$

By successively transforming an initial variable $\xi_0$ with density $p_{\xi_0}$ through a sequence of $K$ invertible mappings $F_k$ as $\xi_K = F_K \circ \cdots \circ F_2 \circ F_1(\xi_0)$, we can derived an arbitrarily flexible density $p_{\xi_K}$ of $\xi_K$ by successively applying equation 11:

$$\log p_{\xi_K}(\xi_K) = \log p_{\xi_0}(\xi_0) - \sum_{k=1}^{K} \log \left| \det \frac{\partial F_k}{\partial \xi_{k-1}} \right|. \tag{12}$$

**Limitations of KL-based functional variational inference**    Note that the Radon-Nikodym derivative between the variational posterior measure and the prior measure is $\frac{\mathrm{d}Q_f}{\mathrm{d}P_0}$, and $\mathrm{KL}[Q_f \| P_0]$ is defined as $\int \log(\frac{\mathrm{d}Q_f}{\mathrm{d}P_0})\mathrm{d}Q_f$ for the infinite-dimensional stochastic processes (Gray, 2011). In the case where $\mathrm{d}Q_f$ is not absolutely continuous with respect to $\mathrm{d}P_0$ will lead to $\mathrm{KL}[Q_f \| P_0] = \infty$ (Matthews et al., 2016; Wild et al., 2022), e.g., the KL divergence between stochastic processes defined by two BNNs with different network structures can be infinite (Ma & Hernández-Lobato, 2021). Moreover, Sun et al. (2019) proved that $\mathrm{KL}[Q_f \| P_0] = \sup_{\mathbf{X} \in \mathcal{X}_\mathbb{N}} \mathrm{KL}[Q_f((f^{\mathbf{X}}) \| P_0(f^{\mathbf{X}})]$, where $\mathcal{X}_\mathbb{N} \doteq \bigcup_{n \in \mathbb{N}} \{ \mathbf{X} \in \mathcal{X}_n \mid \mathcal{X}_n \subseteq \mathbb{R}^{n \times p} \}$ is the set of all finite marginal measurement points in the input domain. However, there is no analytical solution for such a supremum, and even for finite $\mathbf{X}$, this KL term is intractable for many distributions due to the unavailability of the density (Rudner et al., 2022). Therefore, the KL-based functional variational inference is a delicate task that has several limitations in practical applications.

## B    THEORETICAL PROOF

### B.1    PROOF FOR PROPOSITION 3.1

Based on the transformed posterior obtained in the first T-Stage of the main paper for the approximate posterior resulting from the second M-Stage, we derive the approximate posteriors for the following alternating stages in detail as follows:

**V-Stage**    Based on the transformed $F_{\lambda_1^*} \circ Q_{f_{s1}}$, we now can continue to perform the fVI procedure, and the resulting approximate posterior measure is denoted as $F_{\lambda_1^*} \circ Q_{f_{s3}}$ with the corresponding updated posterior over parameters as $q_{s3}(\mathbf{w}_{s3}; \boldsymbol{\theta}_{s3})$.

**M-Stage** Similar to the previous M-Stage, we continue to run the fSGLD from the randomly sampled initial point $\mathbf{w}_{s3} \sim \mathcal{N}(\mu_{s3}, \mathbf{\Sigma}_{s3})$ from the V-Stage, and the corresponding initial function is $F_{\lambda_1^*}(f_{s3}(\cdot; \mathbf{w}_{s3}))$. Then, the probability measure of the collected function samples $F_{\lambda_1^*}(f_{s4_{\{1:N\}}}^{\mathbf{X}_\mathcal{M}})$ is denoted as $F_{\lambda_1^*} \circ Q_{f_{s4}}$ and $F_{\lambda_1^*}(f_{s4}(\cdot; \mathbf{w}_{s4}))$ is the random function mapping at this stage.

**T-Stage** In order to perform the fVI procedure in following V-Stage, it is necessary to fit the second invertible bijection $F_{\lambda_2}^{-1}$ to transform $F_{\lambda_1^*}(f_{s4}(\cdot; \mathbf{w}_{s4}))$ based on $F_{\lambda_1^*}(f_{s3}(\cdot; \mathbf{w}_{s3}))$ obtained from the last V-Stage as $F_{\lambda_1^*}(f_{s4}(\cdot; \mathbf{w}_{s4})) = F_{\lambda_2^*}(F_{\lambda_1^*}(f_{s3}(\cdot; \mathbf{w}_{s3})))$. Then the corresponding transformation for probability measure $F_{\lambda_1^*} \circ Q_{f_{s4}}$ can be represented by $F_{\lambda_2^*} \circ F_{\lambda_1^*} \circ Q_{f_{s3}}$ with parametric $q_{s3}(\mathbf{w}_{s3}; \boldsymbol{\theta}_{s3})$, which is able to run the fVI smoothly and the resulting updated approximate posterior measure is denoted by $F_{\lambda_2^*} \circ F_{\lambda_1^*} \circ Q_{f_{s5}}$ with the corresponding $q_{s5}(\mathbf{w}_{s5}; \boldsymbol{\theta}_{s5})$.

We continue in this fashion by successively alternating between V-Stage and M-Stage, utilizing T-Stage as the intermediary, to create a chain of total $K$ V-Stages and M-Stages. This hybrid approximate process results in a final approximate posterior measure that is in a composite form: $F_{\lambda_{(K-1)/2}^*} \circ \cdots F_{\lambda_2^*} \circ F_{\lambda_1^*} \circ Q_{f_{sK}}$, which is arbitrarily complex and preferably flexible. This compositional structure allows for a more refined posterior approximation, enhancing both its expressiveness and ability to capture complex distributions.

## B.2 PROOF FOR PROPOSITION 3.2

Dynamics-based MCMC sampling techniques are derived from the more general Itô diffusion (Øksendal, 2003), which is often used to characterize the evolution of particles. Given a stochastic process $X : [0, \infty) \times \Omega \to \mathbb{R}^n$ defined on a probability space $(\Omega, \Sigma, P)$, an Itô diffusion in n-dimensional Euclidean space driven by the standard Wiener process satisfies the following specific type of stochastic differential equation (SDE):

$$\mathrm{d}X_t = \mu(X_t)\mathrm{d}t + \sigma(X_t)\mathrm{d}B_t, \tag{13}$$

where $X_t$ is the state of the stochastic process at time $t$, $\mu(\cdot) : \mathbb{R}^n \to \mathbb{R}^n$ is a vector field denoting the deterministic drift term, $\sigma(\cdot) : \mathbb{R}^n \to \mathbb{R}^{n \times n}$ is a matrix field denotes the diffusion coefficient for $X_t$. Both $\mu(\cdot)$ and $\sigma(\cdot)$ are assumed to satisfy the usual Lipschitz continuity condition (Ghosh, 2010). And $B \in \mathbb{R}^n$ is an n-dimensional Wiener process (Brownian motion), $\mathrm{d}B_t$ is the increment of a Wiener process distributed as $\mathrm{d}B_t \sim \mathcal{N}(0, \mathrm{d}t \cdot \mathbf{I}_n)$. Let $p(x, t)$ be the probability density of $X_t$, the evolution of $p(x, t)$ is formed by the following Fokker-Planck (FP) equation (Risken, 1996):

$$\frac{\partial p(x, t)}{\partial t} = -\sum_i \frac{\partial}{\partial x_i} [\mu_i(X_t)p(x, t)] + \sum_{i,j} \frac{\partial^2}{\partial x_i \partial x_j} [D_{i,j}(x, t)p(x, t)], \tag{14}$$

where $D_{i,j}(x, t) = \frac{1}{2}\sigma(X_t)\sigma(X_t)^T$. Ma et al. (2015) proved in Theorem 1 that if $\mu(X_t)$ and $\sigma(X_t)$ are restricted to the following form

$$
\begin{aligned}
\mu(X) &= -[\mathbf{D}(X) + \mathbf{Q}(X)]\nabla U(X) + \Gamma(X), \\
\Gamma(X) &= \sum \frac{\partial}{\partial X}(\mathbf{D}_{ij}(X) + \mathbf{Q}_{ij}(X)), \\
\sigma(X) &= \sqrt{2\mathbf{D}(X)},
\end{aligned} \tag{15}
$$

where $U(\cdot)$ is the potential energy functional, $\mathbf{D}(\cdot)$ is a positive semidefinite matrix, and $\mathbf{Q}(\cdot)$ is a skew-symmetric curl matrix, then the Fokker–Planck equation can be transformed into a more compact form (Yin & Ao, 2006; Shi et al., 2012):

$$\frac{\partial p(x, t)}{\partial t} = \nabla^T \cdot ([\mathbf{D}(X) + \mathbf{Q}(X)][p(x, t)\nabla U(X) + \nabla p(x, t)]). \tag{16}$$

Wu et al. (2024a) proved that their functional Langevin dynamics in Equation (2) can be cast into the above general framework as

$$
\begin{aligned}
\mathrm{d}f_t(\cdot; \mathbf{w}) &= [-(\nabla_{\mathbf{w}}f)^T(\nabla_{\mathbf{w}}f)\nabla_f U(f_t) + H_{\mathbf{w}}f]\mathrm{d}t + \sqrt{2}(\nabla_{\mathbf{w}}f)^T\mathrm{d}B_t \\
&= (-[\mathbf{D}(f) + \mathbf{Q}(f)]\nabla_f U(f_t) + \Gamma(f))\,\mathrm{d}t + \sqrt{2\mathbf{D}(f)}\mathrm{d}B_t,
\end{aligned} \tag{17}
$$

where $U(f) = -\log p(\mathbf{Y}_\mathcal{D}|f(\mathbf{X}_\mathcal{D}; \mathbf{w})) + I_0(f)$ is an OM functional for $P_{f|\mathcal{D}}$, $\mathbf{D}(f) = (\nabla_\mathbf{w} f)^T (\nabla_\mathbf{w} f)$, $\mathbf{Q}(f) = \mathbf{0}$, $\Gamma(f) = H_\mathbf{w} f$. Therefore, the Fokker-Planck equation of the probability measure $p(f, t)$ for $f_t(\cdot; \mathbf{w})$ then can be derived as

$$\frac{\partial p(f, t)}{\partial t} = \nabla^T \cdot ([\mathbf{D}(f) + \mathbf{Q}(f)][p(f, t)\nabla_f U(f) + \nabla p(f, t)]), \tag{18}$$

from which the stationary measure $\pi(f)$ of $p(f, t)$ can be verified as $\exp(-U(f)) = P_{f|\mathcal{D}}$, that is, the target posterior measure over functions.

The proof of Proposition 3.2 is based on the Wasserstein gradient flows, which is an extension version of gradient flows in Euclidean space to the space of probability measures. Formally, suppose $\mathcal{P}(\Omega)$ is the space of probability measures on $\Omega$ with finite second moments and endowed with a Riemannian geometry characterized by the 2-Wasserstein distance (Chen et al., 2018). For a functional $\mathcal{F}$ that maps probability measures to real values: $\mathcal{P}(\Omega) \to \mathbb{R}$, Wasserstein gradient flows describe the evolution of probability measures over time by decreasing the functional $\mathcal{F}$ (Yi & Liu, 2023), e.g., $\mathcal{F}(q)$ can be the $\mathrm{KL}[q\|p]$ for probability measure $q$. Let $(\mathcal{P}(\Omega), W_2)$ be a metric space of $\mathcal{P}(\Omega)$ and be equipped with 2-Wasserstein distance, a curve $\{q_t\} \in \mathcal{P}(\Omega)$ in the Wasserstein space $(\mathcal{P}(\Omega), W_2)$ is the gradient flow of functional $\mathcal{F}$ if it satisfies the following PDE:

$$\frac{\partial q_t}{\partial t} = \mathrm{div}(q_t \nabla_{W_2} \mathcal{F}(q_t)) = \nabla_x \big(q_t \nabla_x (\frac{\delta \mathcal{F}(q_t)}{\delta q_t})\big), \tag{19}$$

where $\nabla_{W_2} \mathcal{F}(q)$ is the Wasserstein gradient of the functional $\mathcal{F}(q)$, $\frac{\delta \mathcal{F}(q)}{\delta q}$ is called the *first variation* of $\mathcal{F}$ at $q$ (Ambrosio et al., 2008) and $\nabla_x$ denotes the Euclidean gradient operation. Let $\{v_t\}$ be a family of vector fields in space $\Omega$ induced by the Wasserstein gradient and it formulates a probability flow ordinary differential equation (ODE) as

$$\mathrm{d}x_t = v_t(x_t)\mathrm{d}t = -\nabla_x(\frac{\delta \mathcal{F}(q_t)}{\delta q_t})\mathrm{d}t, \tag{20}$$

where $v(x) = -\nabla_x(\frac{\delta \mathcal{F}(q)}{\delta q})$. This ODE characterize the evolution of particle $x_t \sim q_t$ in $\Omega$ when the corresponding marginal $q_t$ evolves to decrease $\mathcal{F}(q)$ as in Equation (19).

The Wasserstein gradient flows of parameter-space variational inference for posterior inference is to decrease the following functional $\mathcal{F}$ of the approximate posterior $q(\mathbf{w})$

$$\mathcal{F}(q(\mathbf{w})) \triangleq \underbrace{\int U(\mathbf{w})q(\mathbf{w})\mathrm{d}\mathbf{w}}_{E1} + \underbrace{\int q(\mathbf{w})\log q(\mathbf{w})\mathrm{d}\mathbf{w}}_{E2}$$
$$= \mathrm{KL}[q(\mathbf{w})\|p(\mathbf{w}|\mathcal{D})], \tag{21}$$

where $U(\mathbf{w}) = -\log p(\mathbf{Y}_\mathcal{D}|f(\mathbf{X}_\mathcal{D}; \mathbf{w})) - \log p_0(\mathbf{w})$, $E_2$ is the energy functional of a pure Brownian motion (Zhang et al., 2019). The first variational of $E_1$ and $E_2$ is as $\frac{\delta E_1}{\delta q(\mathbf{w})} = U(\mathbf{w})$ and $\frac{\delta E_2}{\delta q(\mathbf{w})} = \log q(\mathbf{w}) + 1$. Then, the gradient flow of $\mathcal{F}(q(\mathbf{w}))$ is as

$$\frac{\partial q_t(\mathbf{w})}{\partial t} = \nabla_\mathbf{w}\Big[q_t(\mathbf{w})\Big(\nabla_\mathbf{w} U(\mathbf{w}) + \nabla_\mathbf{w} \log q_t(\mathbf{w})\Big)\Big]$$
$$= \nabla_\mathbf{w}\Big[q_t(\mathbf{w})\nabla_\mathbf{w} U(\mathbf{w}) + \nabla_\mathbf{w} q_t(\mathbf{w})\Big], \tag{22}$$

which reads the Fokker–Planck equation of Langevin dynamics for posterior sampling of network parameters as in Equation (16). And the corresponding probability ODE is

$$\mathrm{d}\mathbf{w}_t = -\nabla_\mathbf{w} U(\mathbf{w}_t)\mathrm{d}t - \nabla_\mathbf{w} \log q_t(\mathbf{w}_t)\mathrm{d}t, \tag{23}$$

which share the same marginals $\{q_t(\mathbf{w})\}_{t \geq 0}$ with the Langevin SDE $\mathrm{d}\mathbf{w}_t = -\nabla_\mathbf{w} U(\mathbf{w})\mathrm{d}t + \sqrt{2}\mathrm{d}B_t$ if they evolve from the same $q_0(\mathbf{w})$ (Yi & Liu, 2023).

We now extend this conclusion to the function spaces, that is, to verify whether the functional variation inference and fSGLD share the same marginals. Let $\mathcal{P}(\mathbb{H})$ be the space of Borel probability

measures on function space $\mathbb{H}$ with finite second moments and endowed with a Riemannian geometry. The functional $\mathcal{F}_{fvi}$ for functional variational inference in Equation (1) is

$$\mathcal{F}_{fvi} \triangleq \underbrace{- \int \log p(\mathbf{Y}_{\mathcal{D}} \mid f(\mathbf{X}_{\mathcal{D}}; \mathbf{w})) \mathrm{d}Q_f(f)}_{E1} + \underbrace{\int \log \frac{\mathrm{d}Q_f}{\mathrm{d}P_0}(f) \mathrm{d}Q_f(f)}_{E2} \tag{24}$$
$$= \mathrm{KL}[Q_f \| P_{f|\mathcal{D}}],$$

The Wasserstein gradient flows of the corresponding variational distribution $q(\mathbf{w})$ to this $\mathcal{F}_{fvi}$ is derived as

$$\frac{\partial q_t(\mathbf{w})}{\partial t} = \nabla_{\mathbf{w}} \Big[ q_t(\mathbf{w}) \nabla_{\mathbf{w}} \Big( - \log p(\mathbf{Y}_{\mathcal{D}} \mid f(\mathbf{X}_{\mathcal{D}}; \mathbf{w})) + \log \frac{\mathrm{d}Q_f}{\mathrm{d}P_0}(f) \Big) \Big], \tag{25}$$

and the corresponding probability ODE for $\mathbf{w}_t$ is

$$\mathrm{d}\mathbf{w}_t = \nabla_{\mathbf{w}} \Big[ \log p(\mathbf{Y}_{\mathcal{D}} \mid f_t(\mathbf{X}_{\mathcal{D}}; \mathbf{w})) - \log \frac{\mathrm{d}Q_f}{\mathrm{d}P_0}(f_t) \Big] \mathrm{d}t$$
$$= \nabla_{\mathbf{w}} f \Big[ \nabla_f \Big( \log p(\mathbf{Y}_{\mathcal{D}} | f_t(\mathbf{X}_{\mathcal{D}}; \mathbf{w})) + \log \frac{\mathrm{d}P_0}{\mathrm{d}r_{\mathcal{H},\mathcal{F}}}(f_t) - \log \frac{\mathrm{d}Q_f}{\mathrm{d}r_{\mathcal{H},\mathcal{F}}}(f_t) \Big) \Big] \mathrm{d}t, \tag{26}$$

where $r_{\mathcal{H},\mathcal{F}}$ denotes the corresponding Hausdorff/Riemannian measure, and the second equation holds by applying the chain rule for Radon-Nikodym derivatives (Matthews et al., 2016). It shares the same evolution with the functional Langevin SDE for network parameters as (Wu et al., 2024a)

$$\mathrm{d}\mathbf{w}_t = -\nabla_{\mathbf{w}} U(f_t) \mathrm{d}t + \sqrt{2} \mathrm{d}B_t$$
$$= -\nabla_{\mathbf{w}} f \left[ \nabla_f (- \log p(\mathbf{Y}_{\mathcal{D}} | f_t(\mathbf{X}_{\mathcal{D}}; \mathbf{w})) + I_0 f(t_t)) \right] \mathrm{d}t + \sqrt{2} \mathrm{d}B_t$$
$$= \nabla_{\mathbf{w}} f \left[ \nabla_f (\log p(\mathbf{Y}_{\mathcal{D}} | f_t(\mathbf{X}_{\mathcal{D}}; \mathbf{w})) + \log \frac{\mathrm{d}P_0}{\mathrm{d}r_{\mathcal{H},\mathcal{F}}}(f_t)) \right] \mathrm{d}t + \sqrt{2} \mathrm{d}B_t, \tag{27}$$

where the OM functional $I_0(f)$ for prior measure $P_0$ can be represented by $- \log \frac{\mathrm{d}P_0}{\mathrm{d}r_{\mathcal{H},\mathcal{F}}}(f)$ with $r_{\mathcal{H},\mathcal{F}}$, and the potential energy functional $U(f) = - \log p(\mathbf{Y}_{\mathcal{D}} | f(\mathbf{X}_{\mathcal{D}}; \mathbf{w})) - \log \frac{\mathrm{d}p_0}{\mathrm{d}r_{\mathcal{H},\mathcal{F}}}(f)$. Moreover, the Wasserstein gradient flows in Equation (25) corresponds to the following function-space probability measure evolution (Wang et al., 2019):

$$\frac{\partial Q_f}{\partial t} = \nabla_f \Big[ Q_f (\nabla_{\mathbf{w}} f)^T (\nabla_{\mathbf{w}} f) \nabla_f \Big( - \log p(\mathbf{Y}_{\mathcal{D}} \mid f(\mathbf{X}_{\mathcal{D}}; \mathbf{w})) + \log \frac{\mathrm{d}Q_f}{\mathrm{d}P_0}(f) \Big) \Big]$$
$$= \nabla_f \cdot (\nabla_{\mathbf{w}} f)^T (\nabla_{\mathbf{w}} f) \Big[ Q_f \nabla_f U(f) + \nabla_f Q_f \Big], \tag{28}$$

where the second equation recovers the Fokker-Planck equation in Equation (18) of the probability measure $p(f, t)$ for $f_t(\cdot; \mathbf{w})$ of the functional Langevin SDE in Equation (2). The corresponding probability ODE for $f$ is derived as

$$\mathrm{d}f_t = (\nabla_{\mathbf{w}} f)^T (\nabla_{\mathbf{w}} f) \nabla_f \Big[ -U(f_t) - \log \frac{\mathrm{d}Q_f}{\mathrm{d}r_{\mathcal{H},\mathcal{F}}}(f_t) \Big] \mathrm{d}t, \tag{29}$$

which, therefore, share the same probability measure evolution marginals with functional Langevin SDE in Equation (2). Note that this proposition remains valid even under data subsampling. The justification is twofold: 1) The stochastic gradient approximation of the likelihood term does not alter the target functional of the Wasserstein gradient flow in fVI. This ensures that the overall optimization landscape remains unchanged despite subsampling; 2) It is well-established that the stationary distribution of fSGLD converges to the target posterior when using a decreasing step size. This holds because the stochastic gradient noise decays at a faster rate than the injected Gaussian noise, preserving the asymptotic correctness of the inference process. Thus, the validity of this proposition remains independent of the stochastic gradient approximation, ensuring its robustness even when subsampling is applied. The schematically is shown in the following Figure 3.

## C  PSEUDOCODE FOR FVIMC

Algorithm 1 presents the pseudocode for FVIMC.

---

**Algorithm 1** Functional Variational Inference MCMC (FVIMC)

---

**Require:** Dataset $\mathcal{D} = \{\mathbf{X}_\mathcal{D}, \mathbf{Y}_\mathcal{D}\}$, minibatch $\mathcal{B} = \{x_j, y_j\}_{j=1}^n \subset \mathcal{D}$, functional prior $P_0$, total number of hybrid approximation stages $K$, $U(f) = -\log p(\mathbf{Y}_\mathcal{D}|f(\mathbf{X}_\mathcal{D}; \mathbf{w})) + I_0(f)$

1: Initialise $\mathbf{w}_{s0} \sim \mathcal{N}(\mu_{s0}, \boldsymbol{\Sigma}_{s0})$, reparameterize $\mathbf{w} = \mu + \boldsymbol{\Sigma} \odot \epsilon$ with $\epsilon \sim \mathcal{N}(\mathbf{0}, \mathbf{I})$, $\mu_{s0} \sim \mathcal{N}(\mathbf{0}, \mathbf{I})$, $\boldsymbol{\Sigma}_{s0} \sim \mathcal{N}(\mathbf{0}, \mathbf{I}) + \mathbf{1}$, $\boldsymbol{\theta}_{s0} := \{\mu_{s0}, \boldsymbol{\Sigma}_{s0}\}$

2: **while** $\theta$ not converged **do**

3:    # V-Stage 1:

4:    draw measurement set $\mathbf{X}_\mathcal{M}$ randomly from input domain

5:    draw functional prior functions $f^{\mathbf{X}_\mathcal{M}} \sim P_0$ at $\mathbf{X}_\mathcal{M}$

6:    draw neaural network (NN) functions $f^{\mathbf{X}_\mathcal{M}} \sim Q_{s0}$ at $\mathbf{X}_\mathcal{M}$

7:    $\boldsymbol{\theta}_{s0} \leftarrow \text{Optimizer}(\boldsymbol{\theta}_{s0}, \mathcal{L}_W)$

8:    $\boldsymbol{\theta}_{s1} \leftarrow \boldsymbol{\theta}_{s0}^*$

9:    Obtain $\quad q_{s1}(\mathbf{w}_{s1}; \boldsymbol{\theta}_{s1}), \quad$ corresponding $\quad Q_{f_{s1}} \quad$ and $\quad$ approximated $\mathcal{GP}(f_{s1}|f_{s1}(\cdot, \mu_{s1}), \mathcal{J}_{\mu_{s1}} \boldsymbol{\Sigma}_{s1} \mathcal{J}_{\mu_{s1}}^T)$

10:    # M-Stage 2:

11:    $S \leftarrow \varnothing$;

12:    randomly sample $\mathbf{w}_0 \sim \mathcal{N}(\mu_{s1}, \boldsymbol{\Sigma}_{s1})$

13:    **for** $t = 0$ to $N$ **do**

14:      draw measurement set $\mathbf{X}_\mathcal{M}$;

15:      $\mathbf{w}_{t+1} \leftarrow \mathbf{w}_t - \epsilon_t \nabla_\mathbf{w} \tilde{U}(f_t)) + \sqrt{2\epsilon_t} \eta_t$ using Equation (3);

16:      $S \leftarrow S \cup \{\mathbf{w}_{t+1}\}$;

17:    **end for**

18:    Obtain the implicit $q_{s2}(\mathbf{w}_{s2})$, corresponding $Q_{f_{s2}}$

19:    # T-Stage 1:

20:    Collect corresponding function samples $f^{\mathbf{X}_\mathcal{M}}_{s2_{\{1:N\}}} \sim Q_{f_{s2}}$ on $\mathbf{X}_\mathcal{M}$ based on the network parameters samples $S$

21:    $\lambda_1 \leftarrow \text{Optimizer}(\lambda_1, \mathcal{L})$ in Equation (5) based on the $Q_{f_{s1}}$ from the last V-Stage 1

22:    Obtain the transformed $F_{\lambda_1^*} \circ Q_{f_{s1}}$ for $Q_{f_{s2}}$

23:    # Alternating loops

24:    **for** $i$ in range(3, $K$, 2) **do**

25:      # V-Stage $i$:

26:      draw measurement set $\mathbf{X}_\mathcal{M}$ randomly from input domain

27:      draw functional prior functions $f^{\mathbf{X}_\mathcal{M}} \sim P_0$ at $\mathbf{X}_\mathcal{M}$

28:      draw NN functions $f^{\mathbf{X}_\mathcal{M}} \sim F_{\lambda_{(i-1)/2}^*} \circ \cdots \circ F_{\lambda_1^*} \circ Q_{f_{si-2}}$ at $\mathbf{X}_\mathcal{M}$

29:      $\boldsymbol{\theta}_{si-2} \leftarrow \text{Optimizer}(\boldsymbol{\theta}_{si-2}, \mathcal{L}_W)$

30:      $\boldsymbol{\theta}_{si} \leftarrow \boldsymbol{\theta}_{si-2}^*$

31:      Obain $q_{si}(\mathbf{w}_{si}; \boldsymbol{\theta}_{si})$, corresponding $F_{\lambda_{(i-1)/2}^*} \circ \cdots \circ F_{\lambda_1^*} \circ Q_{f_{si}}$ and approximated $\mathcal{GP}(f_{si}|f_{si}(\cdot, \mu_{si}), \mathcal{J}_{\mu_{si}} \boldsymbol{\Sigma}_{si} \mathcal{J}_{\mu_{si}}^T)$ for $Q_{f_{si}}$

32:      **while** $i + 1 < K$ **do**

33:        # M-Stage $i + 1$:

34:        $S \leftarrow \varnothing$;

35:        randomly sample $\mathbf{w}_0 \sim \mathcal{N}(\mu_{si}, \boldsymbol{\Sigma}_{si})$

36:        **for** $t = 0$ to $N$ **do**

37:          draw measurement set $\mathbf{X}_\mathcal{M}$;

38:          $\mathbf{w}_{t+1} \leftarrow \mathbf{w}_t - \epsilon_t \nabla_\mathbf{w} \tilde{U}(F_{\lambda_{(i-1)/2}^*}(\cdots F_{\lambda_1^*}(f_t))) + \sqrt{2\epsilon_t} \eta_t$ using Equation (3);

39:          $S \leftarrow S \cup \{\mathbf{w}_{t+1}\}$;

40:        **end for**

41:        Obtain the implicit $q_{si+1}(\mathbf{w}_{si+1})$, corresponding $F_{\lambda_{(i-1)/2}^*} \circ \cdots \circ F_{\lambda_1^*} \circ Q_{f_{si+1}}$

42:        # T-Stage $(i + 1)/2$:

43:        Collect corresponding function samples $F_{\lambda_{(i-1)/2}^*}(\cdots (F_{\lambda_1^*}(f^{\mathbf{X}_\mathcal{M}}_{si+1_{\{1:N\}}}))) \sim F_{\lambda_{(i-1)/2}^*} \circ \cdots \circ F_{\lambda_1^*} \circ Q_{f_{si+1}}$ on $\mathbf{X}_\mathcal{M}$ based on the network parameters samples $S$

44:        $\lambda_{(i+1)/2} \leftarrow \text{Optimizer}(\lambda_{(i+1)/2}, \mathcal{L})$ in Equation (5) based on $F_{\lambda_{(i-1)/2}^*} \circ \cdots \circ F_{\lambda_1^*} \circ Q_{f_{si}}$ obtained from the last V-Stage $i$

45:        Obtain the transformed $F_{\lambda_{(i+1)/2}^*} \circ F_{\lambda_{(i-1)/2}^*} \circ \cdots \circ F_{\lambda_1^*} \circ Q_{f_{si}}$ for $F_{\lambda_{(i-1)/2}^*} \circ \cdots \circ F_{\lambda_1^*} \circ Q_{f_{si+1}}$ of the M-Stage $i + 1$

46:      **end while**

47:    **end for**

48: **end while**

---

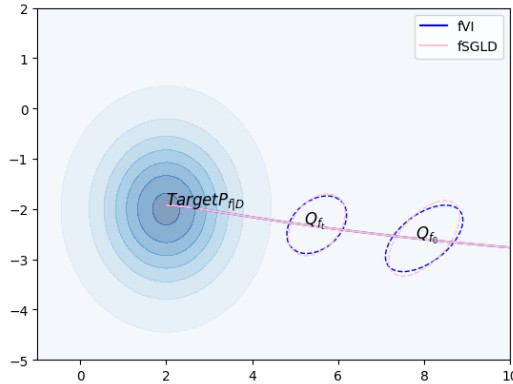

Figure 3: Schematic representation of the same probability measure evolution marginals of fSGLD and the probability ODE derived from the Wasserstein gradient of fVI for posterior inference.

## D  MORE DETAILS FOR INVERTIBLE TRANSFORMATIONS

The specific forms for Linear transformation are as follows:

$$
\begin{aligned}
Forward : y &= \frac{1}{a} * f(x; \mathbf{w}) - \frac{1}{a} * b(x; \omega), \\
Inverse : f(x; \mathbf{w}) &= a * y + b(x; \omega), \\
Jacobian : &\frac{1}{a}
\end{aligned}
\tag{30}
$$

The specific forms for Tanh transformation are as follows:

$$
\begin{aligned}
Forward : y &= a_2 * \text{Tanh}\left(a_1 * f(x; \mathbf{w}) + b_1(x; \omega_1)\right) + b_2(x; \omega_2), \\
Inverse : f(x; \mathbf{w}) &= \frac{1}{a_1} * \left[ \text{arctanh}\left(\frac{y - b_2(x; \omega_2)}{a_2}\right)\right] - \frac{b_1(x; \omega_1)}{a_1}, \\
Jacobian : &a_1 * a_2 * \left[1 - \left(\text{Tanh}(f(x; \mathbf{w}) * a_1 + b_1(x; \omega_1))\right)^2\right]
\end{aligned}
\tag{31}
$$

The specific forms for Sigmoid transformation are as follows:

$$
\begin{aligned}
Forward : y &= a_2 * \text{Sigmoid}\left(a_1 * f(x; \mathbf{w})\right) + b(x; \omega), \\
Inverse : f(x; \mathbf{w}) &= \frac{1}{a_1} * \log\left[\frac{\frac{y - b(x; \omega)}{a_2}}{1 - \frac{y - b(x; \omega)}{a_2}}\right], \\
Jacobian : &a_1 * a_2 * \text{Sigmoid}\left(a_1 * f(x; \mathbf{w})\right) * \left[1 - \text{Sigmoid}\left(a_1 * f(x; \mathbf{w})\right)\right]
\end{aligned}
\tag{32}
$$

where $f(x; \mathbf{w})$ denotes function mappings, $a$ is a scalar parameter and $b(x; \omega)$ denotes a bias function.

## E  EXPERIMENTAL SETTING

**Extrapolation on Synthetic Data**   In this experiment, we employ two-hidden-layer fully connected neural networks with 100 hidden units per layer across all methods. For all functional approaches, the functional GP prior, utilizing an RBF kernel, is pre-trained on 20 training points for 1000 epochs. The bias function used in the bijections during the T-Stage is implemented as a $2 \times 100$ fully connected neural network.

**UCI Regression** In this experiment, we utilize $2 \times 10$ fully connected neural networks for all methods. Using the RBF kernel, the functional GP prior is pre-trained for 100 epochs. The bias function of the bijections in the T-Stage is formed as a $2 \times 100$ fully connected neural network.

**Contextual Bandits** In this experiment, all methods use fully connected neural networks (input-100-100-output). The GP prior (RBF kernel) is pre-trained on 1000 randomly sampled points from training data. All methods are trained using the last 4096 input-output tuples in the training buffer with a batch size of 64 and training frequency of 64 for each iteration.

**Imagine Classification and OOD Detection** In this experiment, we use two-hidden-layer fully connected neural networks with 50 hidden units per layer for all methods. The functional prior is a Dirichlet-based GP designed for classification tasks and is pre-trained on randomly sampled 1000 points for 500 epochs.

## F  COMPLETE INFERENCE PROCESS OF FVIMC

The full inference process of FVIMC for the 1-D extrapolation experiment is shown in the Figure 4 and Figure 5 (since the whole process cannot fit in one figure, we split it into two), where the leftmost column is the posteriors of all V-Stages, the middle column shows the posteriors of all M-Stages, and the rightmost column is the approximation posterior measures in all T-Stages for the transformation of posterior obtained in all M-Stages. The results are organized in 1-20 alternating V-Stage and M-Stage from top to bottom (the final result from the last V-Stage is shown in Figure 1). The final approximate posterior measures is in the composited form of $F_{\lambda_{10}^*} \circ \cdots F_{\lambda_2^*} \circ F_{\lambda_1^*} \circ Q_{f_{21}}$.

## G  FURTHER RESULTS

### G.1  ANALYSIS OF MIXING TIME

The trajectories of fMCMC in the M-Stage of FVIMC, the parameter-space SGLD and the functional fSGLD for the synthetic extrapolation and UCI regression (Yacht) are presented in Figure 6. The model used for the extrapolation experiment is a two-hidden-layer fully connected neural network (100 hidden units in each layer) with 10401-dimensional network parameters. For the UCI regression, the network is $2 \times 10$ fully connected with 191-dimensional parameters. As shown in Figure 6, the top and bottom rows show the results of the toy and UCI experiments, respectively. With the aid of the explicit optimization of fVI in the V-stage, which quickly converges to local high-probability target regions, the fMCMC in the M-stage of our hybrid FVIMC exhibits a rapid mixing rate for high-dimensional parameter space. In the extrapolation example, the fMCMC in the M-stage converges rapidly to the stationary measure within 800 iterations, while fSGLD and SGLD show stabilizing trends until after 2000 iterations. Similarly, for the UCI regression, our FVIMC reaches the stationary in only 400 iterations, while SGLD and fSGLD require more than 500 iterations for mixing.

### G.2  COMPUTATIONAL COMPLEXITY

The running time for all 2000 training iterations of all inference methods in the UCI regression on the small Boston dataset and the large Protein dataset are shown in Table 4. There are 455 training points with 13-dimensional input features for the Boston dataset and 41157 training points with 9-dimensional features in the Protein dataset. For all functional inference methods, the GP prior was pre-trained for 100 epochs, which takes only an extra 1s. In general, parameter-space KLBBB and SGLD are faster than the corresponding functional VI and MCMC methods. In the small Boston dataset, our FVIMC shows similar computational efficiency to the other functional fVI, GWI, IFBNN, and is nearly 5 times faster than FBNN. With the fast convergence of the V-Stage to locally high probabilities, our FVIMC demonstrates its computational efficiency advantage on the larger Protein dataset: the running time of GWI and FBNN is nearly 1.7-2.5× higher than FVIMC. The mixing rate of fSGLD, which performs well on small datasets, has deteriorated at this time and is significantly slower than our hybrid method. Moreover, the convergence processes of

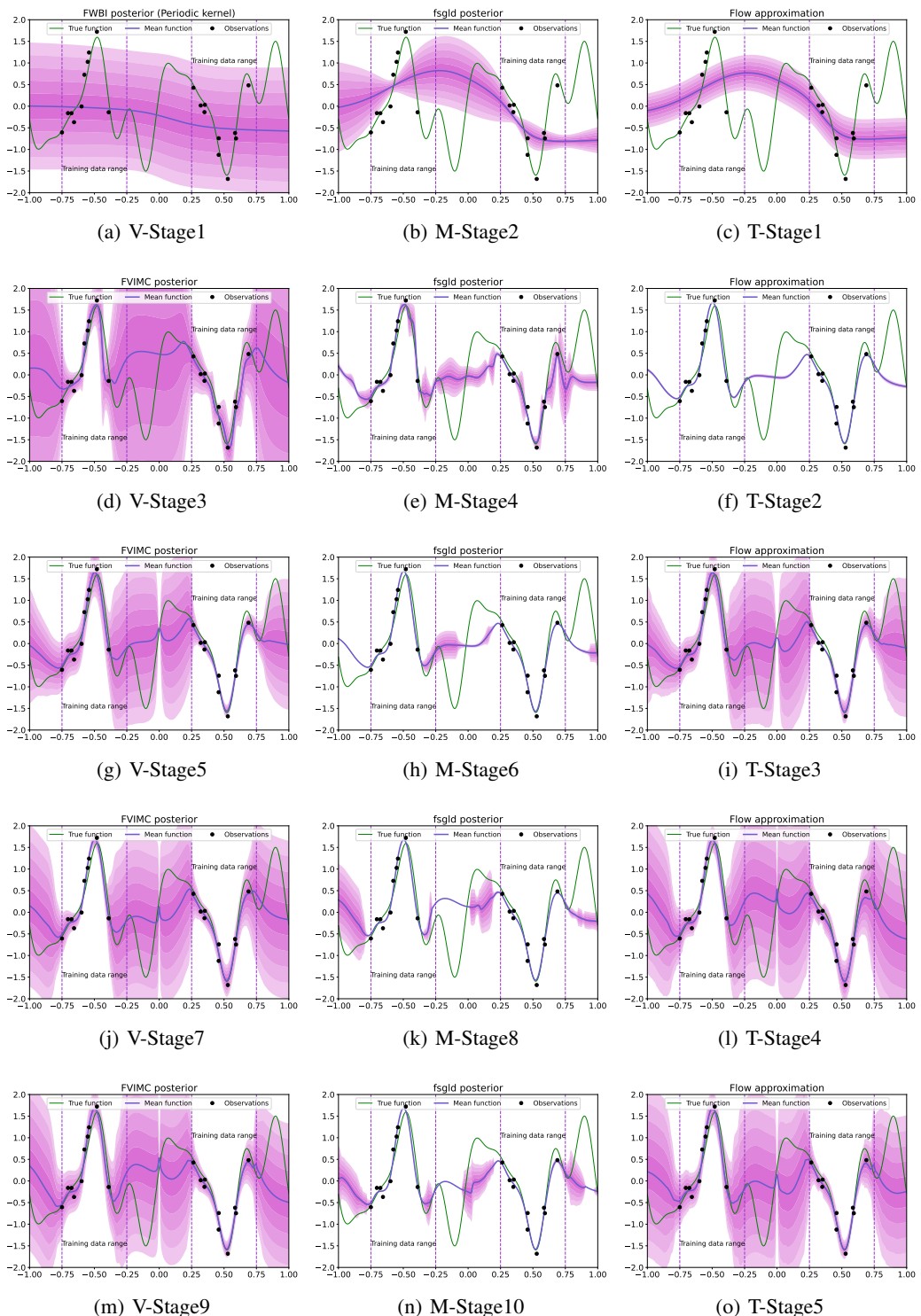

Figure 4: The alternating approximation stages 1-10 in full hybrid inference process of FVIMC.

training loss for all parameter/functiona-space VI methods and the V-Stage of FVIMC are shown in Figure 7, FVIMC achieves high convergence speed and training stability.

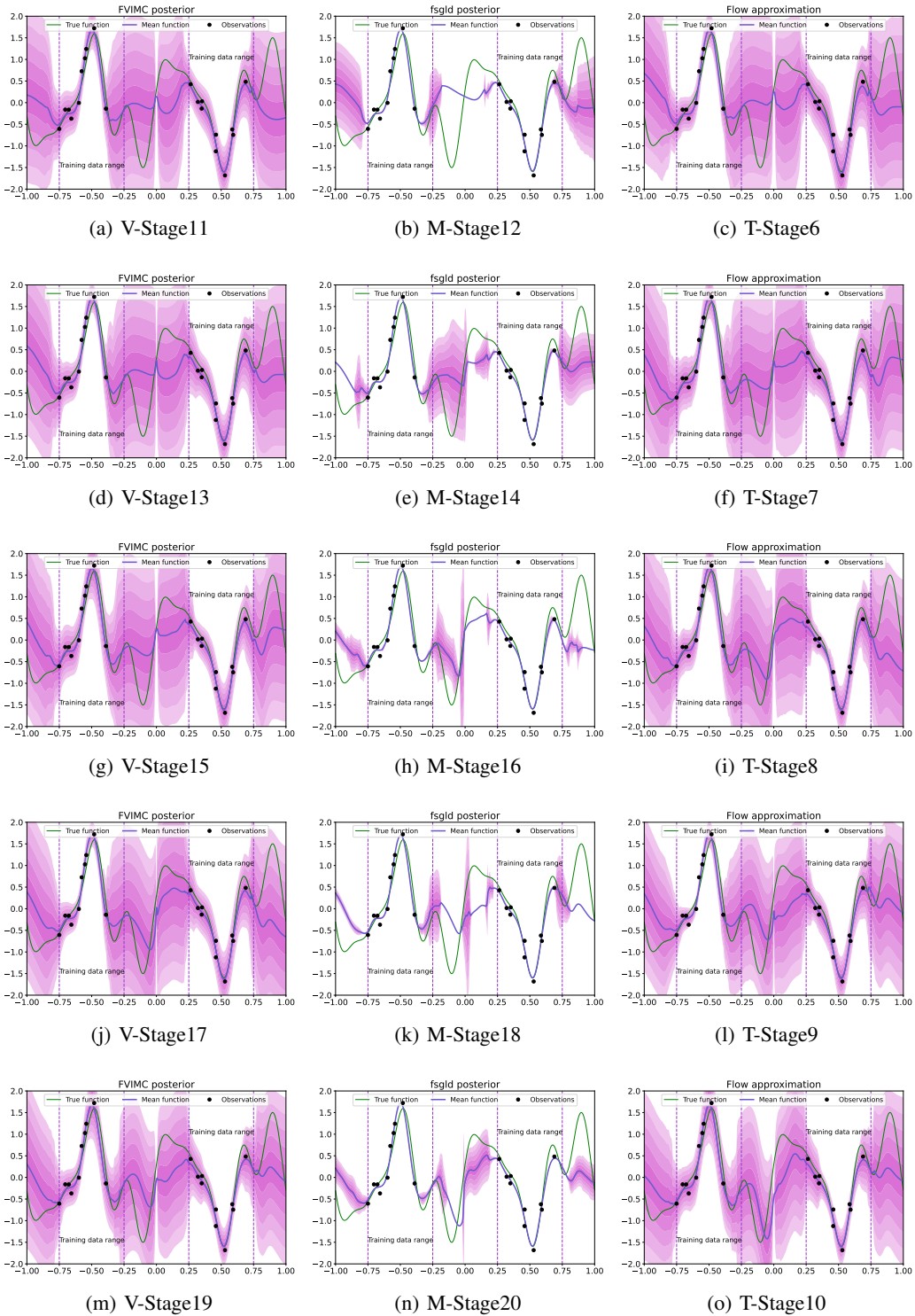

Figure 5: The alternating approximation stages 11-20 in full hybrid inference process of FVIMC.

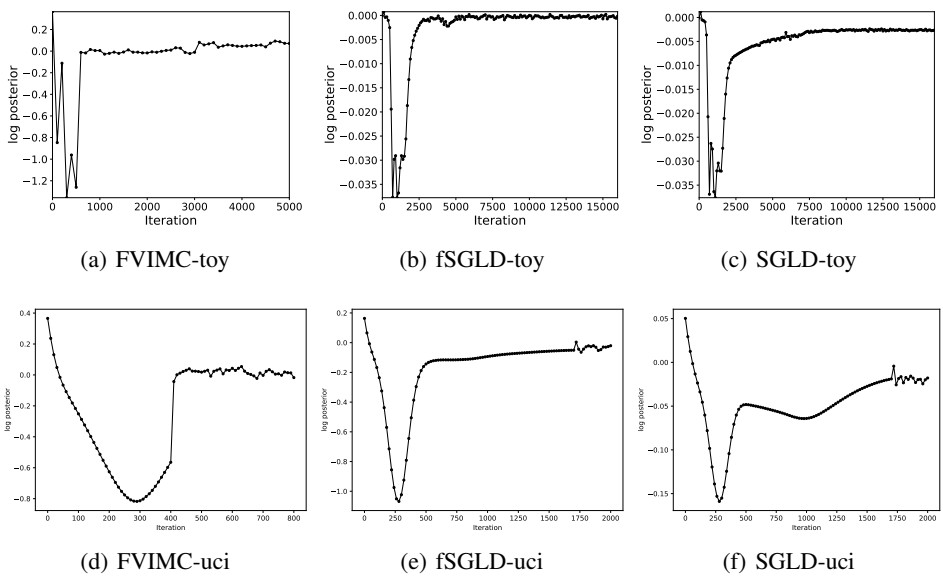

Figure 6: Log-posterior probability versus the number of iterations.

Table 4: Running time comparison on Boston and Protein dataset.

| Run time(s) | Boston | Protein |
|---|---|---|
| FVIMC | $39.000 \pm 0.000$ | $182.500 \pm 0.707$ |
| fVI | $19.000 \pm 0.000$ | $104.000 \pm 2.828$ |
| GWI | $16.000 \pm 1.000$ | $472.667 \pm 0.577$ |
| FBNN | $200.667 \pm 6.531$ | $318.333 \pm 5.774$ |
| IFBNN | $19.000 \pm 0.000$ | $78.500 \pm 2.121$ |
| KLBBB | $8.333 \pm 0.577$ | $8.000 \pm 0.000$ |
| SGLD | $5.500 \pm 0.707$ | $6.500 \pm 2.121$ |
| fSGLD | $9.500 \pm 0.707$ | $239.500 \pm 2.121$ |

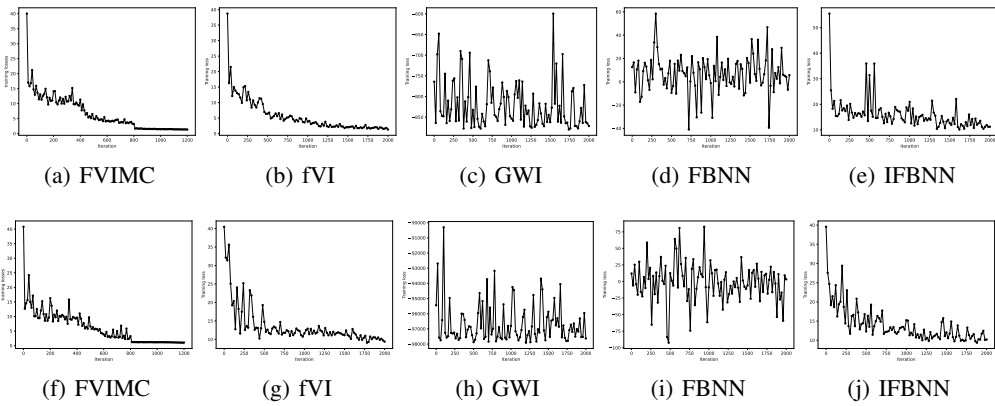

Figure 7: Convergence of training loss for UCI regressions. The top row is the results for the Boston dataset, and the bottom row is the results for the Protein dataset.

## G.3 CALIBRATION CURVES

The calibration curves of all parameter/function-space inference approaches for the synthetic extrapolation example are shown in Figure 8, which assess how well the predicted probabilities of a model match the true frequencies. The horizontal axis represents the predicted cumulative distribu-

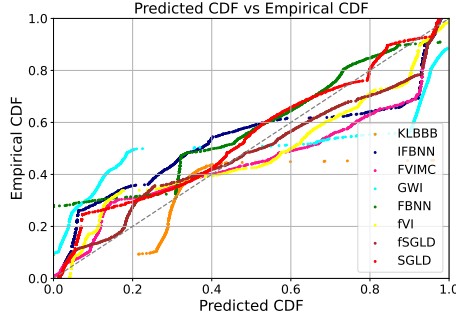

Figure 8: Calibration curves for the synthetic extrapolation. The gray dashed line indicates the perfect calibration.

tion function (CDF), while the vertical axis corresponds to the empirical CDF. The 45-degree dashed diagonal line indicates perfect calibration, where our FVIMC demonstrates competitive calibration capability.

## G.4  Transformation Errors in T-Stage

To quantify the transformation errors introduced in the T-Stage, we plot the predictive RMSE results from the posterior in the M-Stage and the transformed posterior obtained in the T-Stage in the UCI regressions (on Yacht and Boston datasets). As shown in the leftmost column Figure 9(a) and Figure 9(d), the black line represents the RMSE obtained from the M-Stage posterior, the red line denotes the result of the corresponding transformed posterior in the T-Stage, and the blue line is the difference between these two results. With the progress of FVIMC, we can see that the gap between the predicted RMSE by the posterior of the M-Stage and the results corresponding to the approximate posterior obtained through the T-Stage transformation is significantly reduced. Therefore, it can be empirically assumed that this transformation error is controllable, and it will gradually converge to 0 as the whole hybrid inference process proceeds. That is, the parameterized posterior obtained by the T-Satge transformation is a reasonable approximation to the implicit posterior of M-Stage. Additionally, the convergence processes of the training losses in the two T-stages are in the middle and rightmost columns, respectively, and it can be seen that they converge quickly and smoothly.

## G.5  KL Divergence in FVIMC

Empirically, we use the KL divergence-based fVI in the V-Stage to compare the KL-based hybrid FVIMC (denoted as FVIMC(kl)) and our alternative Wasserstein distance-based FVIMC used in the main paper on the UCI regressions. The results are shown in the Table 5, where FBNN represents the KL-based pure functional VI. We can see that under our hybrid inference scheme, FVIMC(kl) outperforms the pure functional variational FBNN, demonstrating the advantage of hybrid inference. However, our FVIMC still achieves superior performance compared to FVIMC(kl), highlighting the benefits of the more flexible Wasserstein distance in capturing complex posterior distributions. Moreover, we present the convergence processes of the KL divergence in FVIMC(kl) and the Wasserstein distance in our FVIMC in Figure 10, where the Wasserstein distance exhibits a significantly more rapid and stable convergence.

Table 5: Comparisons between KL-based FVIMC and Wasserstein distance-based FVIMC

|  |  | FBNN | FVIMC(kl) | FVIMC |
|---|---|---|---|---|
| Yacht | RMSE | 1.52 | 0.92 | 0.37 |
|  | NLL | -0.77 | -0.83 | -5.03 |
| Boston | RMSE | 1.68 | 0.99 | 0.36 |
|  | NLL | -1.19 | -1.62 | -3.74 |

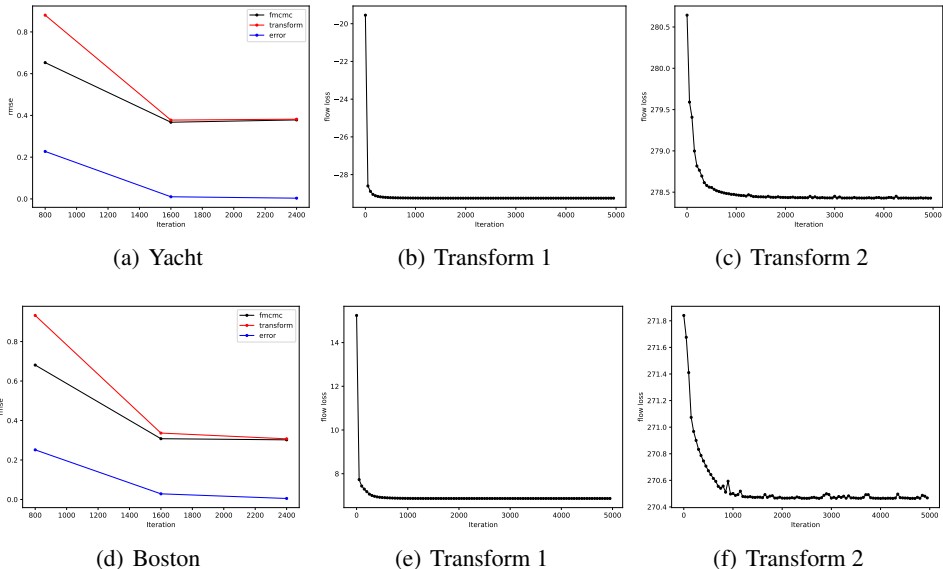

Figure 9: Transformation errors in the T-Stage on UCI regressions. The top row is the result for the Yacht dataset, and the bottom row is the result for the Boston dataset.

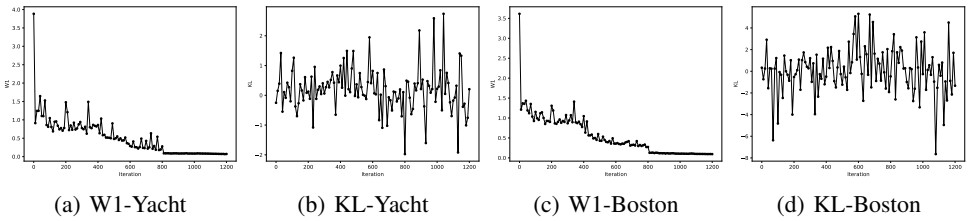

Figure 10: Convergence processes of the KL divergence and the Wasserstein distance in FVIMC, where W1 denotes the Wasserstein distance, and KL denotes the KL divergence.

For the theoretical guarantee of Proposition 3.2, we observe that in some special cases, KL-based FVIMC and Wasserstein-based FVIMC are equivalent. This suggests that replacing the KL divergence with the Wasserstein distance can be interpreted as a relaxation of the original formulation. In the following, we provide a detailed discussion of this relationship.

Let $X$ and $Y$ be two measurable sets, and define $P(X)$ and $P(Y)$ are the sets of all probability measures on X and Y, respectively. Similarly, let $P(X \times Y)$ denote the set of all joint probability measures on $X \times Y$. Given two probability measures (distributions) $q \in P(X)$ and $p \in P(Y)$, the general dual formulation of the 1-Wasserstein distance between $q$ and $p$ is given by $W_1(q, p) := \sup\{\mathbb{E}_q(f) - \mathbb{E}_p(g) : f(x) - g(y) \leq c(x, y)\}$, where $c : X \times Y \to \mathbb{R}$ is a proper distance metric(e.g. Euclidean norm distance), $f : X \to \mathbb{R}$ and $g : Y \to \mathbb{R}$ are real functions, and assume that $X \equiv Y$ (Belavkin, 2018). On the other hand, the KL divergence between $q$ and $p$ can be denoted as $\mathrm{KL}[q\|p] = \mathbb{E}_q(\log q) - \mathbb{E}_q(\log p)$. Interestingly, this can be seen as a special case of the 1-Wasserstein distance when choosing $f = \log q$ and $g = \frac{q}{p} \log p$. Under this condition, the Wasserstein gradient flows of the objective functional $\mathcal{F}_{KL}(q) = \mathrm{KL}[q\|p]$ and $\mathcal{F}_W(q) = W_1(q, p)$ share the same continuity equation as $\frac{\partial q_t}{\partial t} = \mathrm{div}(q_t(\nabla_x \log q_t - \nabla_x \log p))$, which reads the Fokker-Planck equation of the corresponding Langevin SDE $\mathrm{d}x_t = \nabla_x \log p(x_t)\mathrm{d}t + \sqrt{2}\mathrm{d}B_t$. As a result, Proposition 3.2 remains valid for FVIMC when using the Wasserstein distance instead of the KL divergence in this special case. Moreover, to ensure that the 1-Lipschitz continuity condition is met—i.e., $f(x) - g(y) \leq |x - y|$ in the general form of the Wasserstein distance—further considerations and discussions are provided in the following.

Consider first the special case where $q$ and $p$ are of the same type (e.g., both Gaussian distributions with identical means and variances). Under certain local assumptions, it can be shown that the 1-Lipschitz condition $f(x) - g(y) \leq |x - y|$ holds. Since the ratio $q/p$ is constant in this case, the rate of change is controlled. Specifically, at this time, $f = g = \log p$, $f(x) - g(y) = \log p(x) - \log p(y)$. The composition of the logarithmic function with a Gaussian distribution is locally 1-Lipschitz. To satisfy the 1-Lipschitz condition, it should have a bounded first derivative as $|\nabla_x \log p(x)| = \frac{|x-\mu|}{\sigma^2} \leq 1$ by the mean value theorem. Thus, the log-gaussian is 1-Lipschitz in the local range $|x - \mu| \leq \sigma^2$. Additionally, we can verify the derivative condition as $|f'(x) - g'(y)| = |-\frac{x-\mu}{\sigma^2} + \frac{y-\mu}{\sigma^2}| = \frac{|x-y|}{\sigma^2}$. Thus, for $|x - y| \leq \sigma^2$, the function $f(x) - g(y)$ satisfies 1-Lipschitz continuity. This indicates that if the distance between $x$ and $y$ is sufficiently small (bounded by the variance $\sigma^2$), the desired Lipschitz property can be satisfied. For more general cases where $q$ and $p$ are absolutely continuous (differentiable almost everywhere), the Lipschitz condition is influenced by how smoothly $q/p$ varies. If $q/p$ does not exhibit abrupt fluctuations, the behaviour of $g(y)$ remains controlled. Additionally, If $p$ and $q$ are smooth and similar in shape, the variations in $\log q(x)$ and $(q(y)/p(y)) \log p(y)$ will be moderate, further supporting the 1-Lipschitz condition. To enforce the Lipschitz condition in practical applications, a gradient norm regularization term can be added to the dual form of the Wasserstein distance (Gulrajani et al., 2017; Gouk et al., 2021) as $\mathbb{E}_q(f) - \mathbb{E}_p(g) + \lambda(|f'(x) - g'(y)| - 1)^2$, where $f'(x) = q'(x)/q(x)$, $g'(y) = \frac{q'(y) \cdot \log p(y)}{p(y)} + \frac{q(y) \cdot p'(y) \cdot (1 - \log p(y))}{p^2(y)}$, and $\lambda$ is a penalty coefficient.

# H ABLATION STUDY

## H.1 THE EFFECTS OF THE INVERTIBLE TRANSFORMATIONS

We used two different bijection forms as the base for the auto-selection mechanism in the T-Stage for the probability measure transformation in the 1-D extrapolation experiment: the Linear form and the non-linear Tanh model. To explore the effects of the invertible bijections for the transformation of probability measure after each M-Stage and final on the posterior inference, we consider four other bijection schemes: pure Linear transformation, pure Tanh transformation, pure Sigmoid transformation and the likelihood-based auto-selection mechanism based on these three different bijections. The resulting corresponding FVIMC posteriors are shown in Figure 11. For the pure Linear bijection, the prediction of the FVIMC posterior in the middle unseen range $[-0.25, 0.25]$ has been destroyed as in Figure 11(a). The pure Tanh form in Figure 11(b) exhibits some in-between uncertainty quantification pathologies in the non-observation $[-0.25, 0.25]$. And for the pure Sigmoid scheme in Figure 11(c), the uncertainty in the three non-observed regions is underestimated, which can not fully cover the ground truth function. In contrast, the posterior from the auto-selection mechanism in Figure 11(d) significantly outperforms these three mechanisms regarding both predictive accuracy and uncertainty quantification, demonstrating that this log-likelihood maximization-based auto-selection, rather than a pre-fixed form, can approximate the posterior more accurately. The bijections selected by this automatic mechanism are, in order, the following: [Linear, Tanh, Sigmoid, Linear, Linear, Linear, Linear, Linear, Sigmoid, Linear].

## H.2 THE EFFECTS OF THE KERNEL FUNCTION IN GP PRIOR

Functional Gaussian processes (GPs) priors are able to encode prior knowledge about function properties (e.g., periodicity and smoothness) through corresponding kernel functions. We used the RBF kernel in the 1-D extrapolation experiment, which is suited for modelling polynomial functions. We now consider two alternative kernels: the Matern kernel with the smoothness parameter taking the value of $1/2$, which implies that the function will be less smooth; and the Linear kernel, which is not suitable for modelling polynomial oscillatory curves. The results are shown in Figure 12. The top row shows three GP priors with different RBF, Matern and Linear kernels. And the bottom row is the corresponding FVIMC posteriors using these three different GP priors, respectively. Compared to the RBF GP prior in Figure 12(a), the Martern GP prior in Figure 12(b) is slightly less smooth and the variance for the two observation regions $[-0.75, -0.25]$ and $[0.25, 0.75]$ shows some unreasonable expansion. Consequently, the uncertainty estimation of the corresponding FVIMC posterior in Figure 12(e) of these two observation ranges is a little higher, and it can not recover the main trend of the middle unseen region $[-0.25, 0.25]$. The results from the mismatched Linear kernel

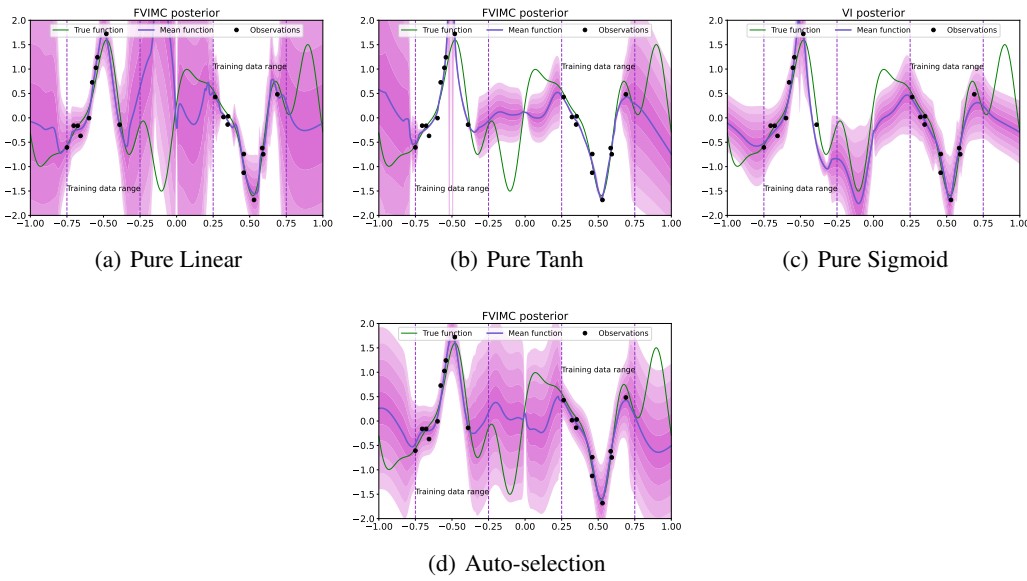

Figure 11: The effects of the bijection forms in the T-Stage.

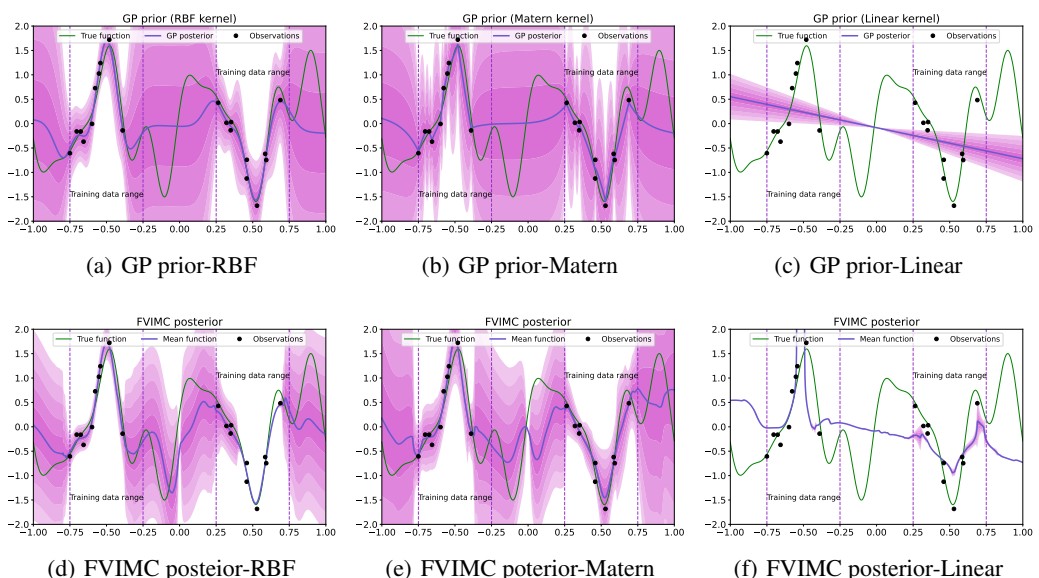

Figure 12: The effects of the kernel function in the functional GP prior. The text after the short line in each subheading represents the corresponding kernel functions.

are shown in Figure 12(c) and Figure 12(f). Since the Linear GP prior shows a completely linear trend and falls to fit the target function, the predictive accuracy and uncertainty estimation of the corresponding FVIMC posterior deteriorates significantly, which indicates that our method can effectively incorporate functional prior information into posterior inference process.

# I NOTATION TABLE

Table 6 is the notation table to demonstrate the notation used in this paper.

Table 6: Notation table

| Notation | Meanings |
|---|---|
| $\mathcal{D} = \{\mathbf{X}_{\mathcal{D}}, \mathbf{Y}_{\mathcal{D}}\}$ | Training dataset |
| $\mathcal{X} \subseteq \mathbb{R}^p$ | ($p$-dimensional) input space |
| $\mathcal{Y} \subseteq \mathbb{R}^c$ | ($c$-dimensional) output space |
| $(\Omega, \mathcal{A}, P)$ | Probability space on $\mathbb{R}^k$ |
| $\mathbb{H}$ | Infinite-dimensional function space (Polish space) |
| $\mathcal{B}(\mathbb{H})$ | Borel $\sigma$-algebra on $\mathbb{H}$ |
| $\mathcal{P}(\mathbb{H})$ | The space of Borel probability measures on $\mathcal{B}(\mathbb{H})$. |
| $\mathbf{X}$ | Finite marginal points |
| $\mathbf{X}_{\mathcal{M}}$ | Finite measurement points |
| $\mathbf{w} \in \mathbb{R}^k$ | Random network parameters |
| $\boldsymbol{\theta} = \{\mu, \boldsymbol{\Sigma}\}$ | Variational parameters |
| $\lambda$ | Parameters of invertible bijections |
| $f(\cdot; \mathbf{w})$ | Random function mapping defined by a BNN parameterized by $\mathbf{w}$ |
| $p_0(\mathbf{w})$ | Prior distribution over network parameters |
| $p(\mathbf{w}\|\mathcal{D})$ | Posterior distribution over network parameters |
| $p(\mathbf{Y}_{\mathcal{D}}\|f(\mathbf{X}_{\mathcal{D}}; \mathbf{w}))$ | Likelihood function evaluated on the training data |
| $q(\mathbf{w}; \boldsymbol{\theta})$ | Variational posterior over network parameters parametrized by $\boldsymbol{\theta}$ |
| $P_0$ | Functional Prior measure |
| $P_{f\|\mathcal{D}}$ | Posterior measure over functions |
| $Q_f$ | Approximate posterior measure over functions |
| $F_\lambda$ | Invertible bijection parametrized by $\lambda$ with inverse $F_\lambda^{-1}$ |

