# OpenReview forum: "Bridging the Gap between Variational Inference and Stochastic Gradient MCMC in Function Space"
_ICLR.cc/2025/Conference — ICLR 2025 Poster_

### Official Review · Reviewer_aL6E · 2024-11-01

**Soundness:** 3
**Presentation:** 3
**Contribution:** 2
**Rating:** 6
**Confidence:** 3

**Summary:**

The paper presents a hybrid functional inference method called Functional Variational Inference MCMC (FVIMC) for posterior approximation. FVIMC alternatives between the V-phase, which optimizes a functional variational objective, and M-phase, which samples a functional Langevin dynamic. The method relies on a learnable transformation to transport the approximate posterior from the M-phase to a density function usable in the V-phase. The paper makes the following claims:
  1. FVIMC is a novel hybrid method that links functional variational inference (FVI) with functional Markov chain Monte Carlo (FMCMC).
  2. FVI and FMCMC share the exact evolution of marginals, which justifies the proposed alternating method.
  3. It provides strong empirical evidence that the proposed method improves predictive performance and uncertainty estimation compared to baselines.

**Strengths:**

- Proposition 3.2 is insightful, novel and sound.
- The evaluation is done against many baseline models, and the UCI results sustain the claim of improved predictive performance.
- The paper suggests a likelihood heuristic for selecting the transformation. The idea is simple and cheap and aligns with an MLE principle.

**Weaknesses:**

## Reason for score
- From the remarks in "Avoid the potential problematic KL divergence for fVI," the objective FVIMC optimizes is not from eq. 2 as the KL divergence is replaced with a Wasserstein-1 distance, and a variance constraint is introduced. It is not clear that proposition 3.2 still applies to FVIMC with these alterations. See question 9.
-  Proposition 3.2 connects FVI and FMCMC when the likelihood is evaluated on $\mathcal{D}$; however, the FVIMC uses data-subsampling. It is unclear whether proposition 3.2 holds with data subsampling. See question 1.
- The proposed method requires running an MCMC chain, optimizing the variational distribution and optimizing a transformation at each iteration. However, the computational cost of their proposed method is not discussed. It would improve the paper to include complexity analysis or empirical evidence comparing runtime and memory overhead to the baselines.
- The method uses several hyperparameters that interact in a complex manner. These include the kernel for the GP prior, the number of MCMC samples $N$, the convergence criteria for the V-phase, and the set of transformations and their architecture. However, the paper does not address how to choose these. It would greatly benefit from guidelines, supported by experiments or analysis, for specifying the hyperparameters.
- The paper does not clearly state that the T-stage preserves the shared evolution of posteriors between the V-stage and M-stage. Including a proposition that this holds would strengthen the claim that proposition 3.2 justifies the validity of FVIMC and would improve the paper's score.
- The extrapolation experiment has an unexplained variance collapse in Fig. 1 at $x=0$ for FVIMC. This is also present in the fig. 7 for the FVIMC posteriors. Notably, this collapse is not present in baseline models. It invalidates the claim that "The hybrid fig1(a) exhibits stronger ability than the pure variational fVI in Figure 1(e), e.g., the smoother fitting curves, and can capture critical features of the non-observed middle range [−0.25, 0.25]," because the collapse point is not on the actual curve. Importantly, there is not even posterior mass covering the curve.

**Questions:**

1. Proposition 3.2 connects FVI and FMCMC when the likelihood is evaluated on $\mathcal{D}$; however, the FVIMC uses data-subsampling. Could you clarify whether proposition 3.2 still holds with data subsampling? If this is the case, are there any constraints on the sampling scheme? If not, please justify that FVIMC still targets the same V- and M-Stage evolution of marginals with subsampling.
  2. The paper states that FVIMC can incorporate meaningful prior information into the posterior. The experiments are done through empirical Bayes, i.e., using a GP learned from data as a prior; however, it is not done with BBB. Why can't you cannot use empirical Bayes to find the prior for BBB? If you can, consider including this to make the comparison on equal footing.
  4. Could you please clarify which pathologies "potential pathologies in deep learning" (line 147) are referring to? The statement as presented is vague and, therefore, hard to judge.
  5. On line 54, the paper calls the isotropic Gaussian prior non-informative. Should this be understood as non-interpretable in the sense introduced by Tran et al. (2022)? If so, consider changing it to non-interpretable instead. Otherwise, please show that an isotropic Gaussian is non-informative for BNNs, like a Jeffreys prior would.
  6. What is the size of the networks used for experimentation? It is only clear that they are two-layered.
  7. At what scale of BNNs can we use FVIMC feasibly? Can you make an inference on a ResNet-sized network (11-25k params)?
  8. What model is used in the image classification and OOD detection experiments? This information should be made available in the article or appendix.
9. From the remarks in "Avoid the potential problematic KL divergence for fVI," the objective variational objective FVIMC optimizes is not from eq. 2 as the KL divergence is replaced with a Wasserstein-1 distance, and a variance constraint is introduced. Does Proposition 3.2 still apply to FVIMC with these alterations? If so, please add a sentence that proposition 3.2 still holds to the Avoid the potential... paragraph. Otherwise, what justifies the validity of FVIMC with the changed objective?
10. Why does the collapse occur in FVIMC Fig. 1a? Are there ways to address or mitigate it?

---

> ### Author Response · Authors · 2024-11-24
>
> We express our gratitude to Reviewer aL6E for positive and insightful feedback. In response to your comments, we prepared a revised version of the manuscript and address specific questions below.
>
> Questions:
>
> $\textbf{Q: }$ Could you clarify whether proposition 3.2 still holds with data subsampling?
>
> $\textbf{Reply: }$ Proposition 3.2 remains valid even with data subsampling. Firstly, the stochastic gradient approximation of the likelihood term does not alter the target functional of the Wasserstein gradient flow in fVI. Secondly, it is well-established that the stationary distribution of stochastic gradient Langevin dynamics converges to the target posterior when using a decreasing step size, as the stochastic gradient noise diminishes faster than the injected Gaussian noise. Hence, the validity of Proposition 3.2 is independent of the stochastic gradient approximation. We will include a note on this in the final version.
>
> $\textbf{Q: }$ The paper states that FVIMC can incorporate meaningful prior information into the posterior. The experiments are done through empirical Bayes, i.e., using a GP learned from data as a prior; however, it is not done with BBB. Why can't you cannot use empirical Bayes to find the prior for BBB?
>
> $\textbf{Reply: }$ The reason is that functional prior like GP can easily encode prior knowledge about the underlying functions, e.g., periodicity, shape and smoothness through corresponding kernel functions. For example, suppose we want to predict the daily average temperature of a country. We know that such a function should show some periodicity, so we can give GP prior with a periodicity kernel, which can help with function learning. In contrast, it is hard to encode such periodicity into each neural network weight. As network weights are not interpretable, formulating a good prior on them is virtually impossible.
>
> $\textbf{Q: }$ Could you please clarify which pathologies "potential pathologies in deep learning" (line 147) are referring to?
>
> $\textbf{Reply: }$ We apologize for not elaborating on these pathologies due to space limitations. There is existing literature has highlighted potential issues with isotropic Gaussian priors in parameter-space inference. For instance, as network depth increases, function samples derived from Gaussian priors over parameters tend to exhibit horizontal behavior [1][2]. Moreover, in architectures with ReLU activations, the prior distribution of unit outputs becomes increasingly heavy-tailed with greater depth. These pathologies in parameter-space priors may lead to problems in posterior inference. Refer to Fig. 1 in [3] for a more intuitive illustration.
>
> [1] David Duvenaud et al. Avoiding pathologies in very deep networks. AISTATS, 2014.
>
> [2] Matthews et al. Gaussian process behaviour in wide deep neural networks. ICLR, 2018.
>
> [3] Tran et al. All you need is a good functional prior for Bayesian deep learning. JMLR, 23(74):1–56, 2022.
>
> $\textbf{Q: }$ On line 54, the paper calls the isotropic Gaussian prior non-informative. Should this be understood as non-interpretable in the sense introduced by Tran et al. (2022)?
>
> $\textbf{Reply: }$ Actually, the default isotropic Gaussian prior is commonly thought of as uninformative and may even carry incorrect beliefs about the posterior (uni-modality, independent weights, etc.[4][5]). Moreover, due to the complex architecture and non-linearity of BNN, the corresponding prior over functions induced by this prior over parameters is also non-interpretable and hard to control, as shown in  Tran et al. (2022).
>
> [4] Knoblauch et al.Generalized variational inference: Three arguments for deriving new posteriors, 2019.
>
> [5] Vincent Fortuin. Priors in Bayesian deep learning: A review, 2022.
>
> $\textbf{Q: }$ What is the size of the networks used for experimentation?
>
> $\textbf{Reply: }$ In the synthetic extrapolation, we use a $2 \times 100$ fully connected tanh neural network for all inference methods. In the uci regressions, the model is a two-hidden-layer fully connected neural network, with each layer having 10 hidden units. In the contextual bandit, we use fully connected neural networks with input-100-100-output architecture. And for the image classification and OOD detection experiments, the model is a two-hidden-layer fully connected neural network, each with 50 hidden units. We add the details of the experiment setting in Appendix E.

---

> > ### Comment · Reviewer_aL6E · 2024-11-25
> > **Responds 1**
> >
> > ## R1
> >
> > Thank you for the clarification. Regarding the SGLD, I assume the authors are referencing the argument in [1] that supports the omission of the Metropolis-Hastings (MH) step.
> >
> > ## R2
> > I understand that interpreting the prior knowledge encoded with an empirical prior for BBB may not be straightforward. However, I would have anticipated better results on UCI datasets when using an empirical prior.
> >
> > Could you elaborate on how you determined the bandwidth for the RBF kernel used in the functional prior?
> >
> > ## R3
> > This point seems to address the functional prior rather than your proposed method. I overlooked this detail in my initial review of your paper, but I now understand and appreciate why you chose not to include it.
> >
> > ## R4
> >
> > There seems to be a potential clash of terminology between communities here. When I think of an uninformative prior, I associate it with a Jeffreys prior invariant to parameterization. In contrast, I would classify a Gaussian prior as informative, reflecting underlying beliefs, even if they are not explicitly stated. I now understand what the authors intended to convey from the clarification provided.
> >
> > ## R5
> >
> > Thanks for adding this information. It provides a sense of the scale at which the method operates.
> >
> > ## References
> > Welling, Max, and Yee W. Teh. "Bayesian learning via stochastic gradient Langevin dynamics." Proceedings of the 28th international conference on machine learning (ICML-11). 2011.

---

> ### Author Response · Authors · 2024-11-24
>
> $\textbf{Q: }$ At what scale of BNNs can we use FVIMC feasibly? Can you make an inference on a ResNet-sized network (11-25k params)?
>
> $\textbf{Reply: }$ The network parameter is 10,401-dimensional and 261-dimensional (maximum) for the synthetic extrapolation and uci regressions (maximum), respectively. The params for the image classifications is 42,310-dimensional, and for the contextual bandit, it is 12,501-dimensional, which illustrates that our FVIMC can scale to large models (at least 10-50k params). Hence, it should be feasible to apply it to ResNet-sized networks. We will consider deploying FVIMC to more complex architectures such as CNN and RNN in our future work.
>
> $\textbf{Q: }$ Why does the collapse occur in FVIMC Fig. 1a? Are there ways to address or mitigate it?
>
> $\textbf{Reply: }$ We consider this to be the result of a minor program implementation bug, as it happens to be near the 0 point. The problem is definitely controllable, and we will double-check and debug our code to solve it.
>
> $\textbf{Q: }$ The computational cost of their method.
>
> $\textbf{Reply: }$ Thanks for the suggestion! We add the analysis of computational complexity in Appendix G.2. To compare the efficiency between FVIMC and other inference approaches, we provide running time for all 2000 training epochs of all inference methods in the uci regression on the small Boston dataset(455 training points) and the large Protein dataset(41157 training points) in Tab4 in Appendix G.2. In the small Boston dataset, our FVIMC shows similar computational efficiency to the other functional fVI, GWI, IFBNN, and is nearly 5 times faster than FBNN. With the fast convergence of the V-Stage to locally high probabilities, our FVIMC demonstrates its computational efficiency advantage on the larger Protein dataset: the running time of GWI and FBNN is nearly 1.7-2.5× higher than FVIMC. The mixing rate of fSGLD, which performs well on small datasets, has deteriorated at this time and is significantly slower than our hybrid method. Moreover, the convergence processes of training loss for all VI baselines and the V-Stage of FVIMC are shown in Fig7, where FVIMC achieves high convergence speed and training stability.
>
> $\textbf{Q: }$ How to choose hyperparameters?
>
> $\textbf{Reply: }$ Our experimental tasks align with the benchmark tasks commonly used in the literature on functional and parameter posterior inference for BNNs. Initially, we referred to the general experimental settings reported in prior studies to define a broad range of hyperparameters. Through pre-training or initial trials, we selected specific configurations that enable all baseline methods to achieve satisfactory results. For instance, we employed an RBF kernel GP for all functional methods to facilitate smoothing, as it is a versatile kernel capable of capturing the primary characteristics of most polynomial functions. Additionally, an extensive ablation study on the choice of kernel functions is provided in Appendix H.2. Regarding convergence criteria for the V-Step and M-Step, we relied on the training loss convergence curve for fVI and the mixing time trajectory for fMCMC, respectively, as detailed in Appendices G.1 and G.3. For the transformation, we utilized a bias function $b(x;w)$, modelled as a two-hidden-layer fully connected neural network with 100 hidden units per layer. This architecture is flexible and expressive, making it well-suited for fitting complex posterior distributions.
>
> $\textbf{Q: }$  State the T-Stage preserves the shared evolution of posteriors between the V-Stage and M-Satge.
>
> $\textbf{Reply: }$ Thanks for the suggestion! We add the plot of the rmse comparisons between the posterior in the M-Stage and the transformed posterior obtained in the T-Stage in the uci regressions to quantify the transformation errors empirically in Appendix G.4. We can see that as the algorithm proceeds, this transformation error can be significantly reduced and effectively controlled. Additionally, the training losses in the T-stages converge quickly and smoothly. Therefore, the parameterized posterior obtained by the T-Satge transformation is a reasonable approximation to the implicit posterior of M-Stage.

---

> ### Comment · Reviewer_aL6E · 2024-11-25
> **Responds 2**
>
> With the author's rebuttal and the updated article, the authors have adequately addressed most of my concerns with the paper. Therefore, I have increased the score to a 5, Soundness to 2 and the Presentation to a 3.  To improve the score to a 6, I would like the authors to comment on question 8.

---

> > ### Author Response · Authors · 2024-11-29
> > **Further reply**
> >
> > Thank you very much for your reply and valuable comments! For the bandwidth, we pre-trained the functional GP to obtain a proper bandwidth for the RBF kernel. It is pre-trained on the training data (or its randomly sampled subset) only for 1000, 100, 100 and 500 epochs for the synthetic extrapolation, uci regressions, contextual bandit and image classification, respectively, which added only an extra 1s, 1s, 1s, and 3s, so it does not cause any significant computational burden.
> >
> > For question 8, we found that KL-based FVIMC and Wasserstein-based FVIMC are the same in a special case. So this Wasserstein-based replacement might be seen as a relaxation of the original KL-based one. The detailed discussions are as follows.
> >
> > Let $X$ and $Y$ be two measurable sets, and let $P(X)$ and $P(Y)$ be the sets of all probability measures on $X$ and $Y$, respectively, and let $P(X \times Y)$ be the set of all joint probability measures on $X \times Y$. Given two probability measures (distributions) $q \in P(X)$ and  $p \in P(Y)$, consider the general dual formation of the 1-Wasserstein distance[1][2] between $q$ and $p$ as $W_1(q, p) := \sup \\{E_{q}(f) - E_{p}(g) : f(x) - g(y) \leq c(x, y) \\}$, where $c: X \times Y \rightarrow \mathbb{R}$ is a proper distance metric(e.g. Euclidean norm distance), $f: X \rightarrow \mathbb{R}$ and $g: Y \rightarrow \mathbb{R}$ are real functions, and assume that $X  \equiv Y$. On the other hand, the KL divergence between $q$ and $p$ can be denoted as $\mathrm{KL}[q||p] = E_{q}(\log q) - E_{q}(\log p)$, which can be regarded as a special case of the 1-Wasserstein distance when $f = \log q$ and $g = \frac{q}{p}\log p$. In this special case, the Wasserstein gradient flows of these two objective functional $F_{\mathrm{KL}}(q) = \mathrm{KL}[q||p]$ and  $F_{W}(q) = W_1(q, p)$ share the same continuity equation as $\frac{\partial q_{t}}{\partial t} = \operatorname{div} (q_{t}(\nabla_{x} \log q_{t} - \nabla_{x} \log p))$, which reads the Fokker-Planck equation of the corresponding Langevin SDE $\mathrm{d} x_{t} = \nabla_{x} \log p(x_{t}) \mathrm{d}t + \sqrt{2} \mathrm{d}B_{t}$. Therefore, Prop 3.2 still applies to FVIMC with the alternative Wasserstein distance in this special case.
> >
> > Empirically, we compared this replacement on UCI regression tasks, where KL-based fVI and Wasserstein-based fVI were used in the V-stage of FVIMC, respectively. The results are shown in the following table, where FBNN represents the KL-based pure functional VI, FVIMC(kl) denotes the KL-based FVIMC.
> >
> > |  dataset   |  metric   |    FBNN   |   FVIMC(kl)    |   FVIMC    | \\\
> > |    yacht    |    rmse   |      1.52    |    0.92        |      0.37        |   \\\
> > |   --------     |    nll       |      -0.77  |      -0.83      |      -5.03        |   \\\
> > |    boston    |    rmse   |      1.68    |    0.99        |      0.36        |   \\\
> > |   --------     |    nll    |      -1.19  |      -1.62     |      -3.74       |
> >
> > We can see that under our hybrid inference scheme, KL-based FVIMC outperforms the pure fVI, demonstrating the advantage of hybrid inference. Moreover, our Wassertein-based FVIMC achieves superior performance compared to KL-based FVIMC, highlighting the benefits of the more flexible Wasserstein distance in capturing complex posterior distributions. We add the plot of the convergence curves of the KL divergence and the Wasserstein distance in Fig. 10 in Appendix G.5, where the Wasserstein distance exhibits a significantly more rapid and stable convergence.
> >
> > [1] Leonid Vitaliyevich Kantorovich. On the transfer of masses. Doklady Akademii Nauk SSSR, 1942.
> >
> > [2] Roman V Belavkin. Relation between the kantorovich–wasserstein metric and the kullback–leibler divergence. In Information Geometry and Its Applications, 2018.

---

> ### Comment · Reviewer_aL6E · 2024-11-29
> **response**
>
> Thanks for sharing the connection between KL and $W_1$. I follow your argument, and your empirical evidence is persuasive. However, I have one point of technical clarification: for KL and $W_1$ to agree, you’d need the difference function f - g to be 1-Lipchitz; otherwise, it’s not a test function. Is this true for $f = \log q$ and g = $\frac{q}{p} \log p$?

---

> > ### Author Response · Authors · 2024-12-01
> > **Further reply**
> >
> > Thank you very much for your response and valuable comments! Consider first a special case when $q$ and $p$ are of the same type (e.g., both Gaussian distributions with identical means and variances); it can be shown that under some local assumptions, $f(x) - g(y) \leq |x - y|$. In this case, the ratio $q/p$ is constant, and the rate of change is controlled. Specifically, at this time, $f = g = \log p$, $f(x) - g(y) = \log p(x) - \log p(y)$. And the composition of the logarithmic function and gaussian is locally 1-Lipschitz. Specifically, to satisfy the 1-Lipschitz condition, it should have a bounded first derivative as $|\nabla_{x} \log p(x)| = \frac{|x - \mu|}{\sigma^{2}} \leq 1$ by the mean value theorem. Thus, the log-gaussian is 1-Lipschitz in the local range $|x - \mu| \leq \sigma^{2}$. Also we can check the derivative condition as  $|f'(x) - g'(y)| =  |-\frac{x - \mu}{\sigma^{2}} + \frac{y - \mu}{\sigma^{2}}| = \frac{|x - y|}{\sigma^{2}}$, so when $|x - y| \leq \sigma^{2}$, $f(x) - g(y)$ satisfy 1-Lipschitz continuity. This indicates that if the distance between $x$ and $y$ is small (limited by the variance $\sigma^{2}$), the desired Lipschitz property can be satisfied.
> >
> > For more general cases, suppose $q$ and $p$ are absolutely continuous and thus differentiable almost everywhere; if $q/p$ changes smoothly without dramatic fluctuations, the behaviour of $g(y)$ will be controlled. Also, suppose $p$ and $q$ are smooth and close in shape, then the changes in $\log q(x)$ and $(q(y)/p(y)) \log p(y)$ will be moderate, supporting the 1-Lipschitz condition.  In a practical implementation, we can guarantee that this Lipschitz condition holds by adding the gradient norm as a regularization term to the dual form [1][2] as $E_{q}(f) - E_{p}(g) + \lambda (|f'(x) - g'(y)| -1)^{2}$, where $f'(x) = q'(x)/q(x)$, $g'(y)=\frac{q'(y) \cdot \log p(y)}{p(y)}+\frac{q(y) \cdot p'(y) \cdot(1-\log p(y))}{p^{2}(y)}$, and $\lambda$ is a penalty coefficient.
> >
> > [1] Gulrajani et al. Improved Training of Wasserstein GANs. NIPS, 2017.
> >
> > [2] Gouk et al, Regularisation of neural networks by enforcing Lipschitz continuity, JML, 2020

---

> > > ### Comment · Reviewer_aL6E · 2024-12-01
> > >
> > > Thanks for clarifying this last point. I have increased the score to 6 and soundness to 3.

---

> > > > ### Author Response · Authors · 2024-12-02
> > > > **Thank you！**
> > > >
> > > > Dear Reviewer aL6E,
> > > >
> > > > We want to express our sincere gratitude for your invaluable feedback on our manuscript. Your insightful comments and suggestions have greatly improved the quality of our work, and we deeply appreciate the time and effort you dedicated to reviewing it.
> > > >
> > > > Once again, thank you very much for your constructive feedback and the increasing score.
> > > >
> > > > Best Regards,
> > > >
> > > > Authors

---

### Official Review · Reviewer_qXT6 · 2024-11-01

**Soundness:** 3
**Presentation:** 4
**Contribution:** 3
**Rating:** 6
**Confidence:** 3

**Summary:**

This paper proposes a novel hybrid posterior inference method that combines the advantages of both functional variational inference and functional MCMC. To frame these two update schemes as an alternating approximation process, the authors aim to learn a bijective transformation of the variational distribution updates by maximizing the log-likelihood functions of the samples collected from functional MCMC steps.
Moreover, since using KL divergence in functional variational inference (fVI) may be potentially problematic, the authors propose replacing it with the Wasserstein distance in their method, fVIMC.
In experiments, the authors validate their method on several benchmark tasks, including the Bayesian neural networks on UCI datasets, exploration-exploitation performance in contextual bandits and classification tasks on MNIST and Fashion MNIST datasets. The results indicate that fVIMC outperforms the primary baseline, like fVI and fMCMC.

**Strengths:**

- The paper proposes an interesting framework that combines variational inference and MCMC in functional space.
- The details of toy experiments are well-explained, and the real-data tasks are comprehensive. The discussion of related work is also very clear.

**Weaknesses:**

The fVIMC method proposed in this paper is natural and interesting, but I have some specific concerns about fVIMC:
- The mechanism of using the 1-Wasserstein distance to replace KL divergence in functional priors seems appropriate. However, this may cause the variational distribution update in fVI step no longer corresponding to the gradient flow of the functional $KL[q(w)|p(w|D)]$. In that case, would the result in Proposition 3.2 still hold?
- In the results for BNNs, the RMSE scale seems inconsistent with previous BNN results reported in [1] and [2].
- In the experimental results, an ablation study comparing KL divergence and 1-Wasserstein distance in fVIMC would be helpful.


[1] Liu, Qiang, and Dilin Wang. "Stein variational gradient descent: A general purpose bayesian inference algorithm." Advances in neural information processing systems 29 (2016).

[2] Sun, Shengyang, et al. "Functional variational Bayesian neural networks." ICLR 2019.

**Questions:**

- Typo: 1.
In line 132, the term $\mathcal{X}^n = \bigcup \\{\mathbf{X} \in \mathcal{X}^n \mid \mathcal{X}^n \in \mathbb{R}^n \\}$ is confusing.

- In line 166, "this factorized Gaussian will be modified later rather than being fixed, as in fVI, so it does not impose a strict restriction." I believe that the choice of variational distribution family $q_{s0}(\mathbf{w}_{s0},\boldsymbol{\theta}_{s0})$ does, in fact, constitute a restriction for fVI. Can you clarify this point more clearly?

---

> ### Author Response · Authors · 2024-11-24
>
> We express our gratitude to Reviewer qXT6 for positive and insightful feedback. In response to your comments, we prepared a revised version of the manuscript and address specific questions below.
>
> Questions:
>
> $\textbf{Q: }$ Typo: 1. In line 132, the term $\mathcal{X}^n = \cup [\mathbf{X} \in \mathcal{X}^n |\mathcal{X}^n \in \mathbb{R}^n ]$ is confusing.
>
> $\textbf{Reply: }$ We are sorry for this typo. Actually, it should be $\mathcal{X}_{\mathbb{N}}  \doteq  \cup _{n \in \mathbb{N}} [\mathbf{X} \in \mathcal{X}_n \mid \mathcal{X}_n \subseteq \mathbb{R}^{n \times p}]$, which denotes the collection of all finite sets of marginal measurement points in the input domain and $p$ is the input feature dimensions.
>
> $\textbf{Q: }$ In line 166, "this factorized Gaussian will be modified later rather than being fixed, as in fVI, so it does not impose a strict restriction." I believe that the choice of variational distribution family $q_{s0}(\mathbf{w}{s0},\boldsymbol{\theta}{s0})$ does, in fact, constitute a restriction for fVI. Can you clarify this point more clearly?
>
> $\textbf{Reply: }$ We understand that our earlier explanation might have been unclear, and we apologise for any confusion. The key point we intended to convey is this: Unlike the naive pure fVI, which typically imposes rigid distributional assumptions on the variational posterior, our hybrid algorithm is inherently more flexible, allowing it to model arbitrarily complex posteriors. Specifically, given a simple initial approximate posterior over parameters  $q_{s0}(\mathbf{w}{s0},\boldsymbol{\theta}{s0})$, e.g., it can be a factorized Gaussian has $\boldsymbol{\theta}{s0} = (\mathbf{\mu}{s0}, \mathbf{\Sigma}{s0})$, which induces the posterior measure over functions denoted by $Q_{f_{s0}}$. Then, after the first V-Stage, we can obtain the updated functional posterior $Q_{f_{s1}}$ with the corresponding updated variational posterior over parameters $q_{s1}(\mathbf{w}{s0}; \boldsymbol{\theta}{s1})$. In the following M-Stage, the probability measure of the collected function samples is denoted as $Q_{f_{s2}}$. Notice that at this point, the distribution $q_{s2}(\mathbf{w}{s2})$ of the corresponding parameter samples becomes assumption-free and implicit, meaning it is no longer restricted to the form of a fully-factorized Gaussian. In contrast, pure fVI retains its initial assumption about the family of variational distributions throughout the optimization process. For example, if the initial variational distribution $q(\mathbf{w},\boldsymbol{\theta})$ is a fully-factorized Gaussian, pure fVI continuously optimizes the variational parameters $\boldsymbol{\theta} = (\mathbf{\mu}, \mathbf{\Sigma})$ within this framework, without altering the underlying distributional type. This oversimplified variational assumption in pure fVI may fall short in accurately approximating the posterior for complex models and induce pathologies such as over-pruning.
>
> Weaknesses:
>
> $\textbf{W: }$ In the results for BNNs, the RMSE scale seems inconsistent with previous BNN results reported in [1] and [2].
>
> $\textbf{Reply: }$ Our experimental setup is different from Sun et al. In their uci experiment, they used different network architectures and training epochs, but the model implementation exactly follows their published code. The results of all methods reported in our paper share the same architecture and training iterations for fair comparison.
>
> $\textbf{W: }$ In the experimental results, an ablation study comparing KL divergence and 1-Wasserstein distance in fVIMC would be helpful.
>
> $\textbf{Reply: }$ In the experiment part of the main paper (Sec.5), FBNN (Sun et al., 2019) refers to the KL divergence-based functional variational method, while fVI represents the alternative function variational inference approach utilizing the Wasserstein distance. Experimental results demonstrate that the pure fVI method, leveraging the Wasserstein distance, outperforms FBNN in both predictive performance and uncertainty estimation. Moreover, KL-based FBNN was excluded from the V-Stage in our hybrid FVIMC for the ablation study due to its prohibitively high computational complexity (you can see Appendix G.2 for detailed computational complexity comparisons), which limits its scalability to models with high-dimensional inputs or targets.

---

> > ### Comment · Reviewer_qXT6 · 2024-11-29
> >
> > Thanks for the response. I will maintain my evaluation.

---

> > > ### Author Response · Authors · 2024-11-29
> > > **Thank you!**
> > >
> > > Dear Reviewer qXT6,
> > >
> > > Thank you very much for taking the time to review our responses. We hope we have addressed your questions and concerns. Please feel free to let us know if there are any remaining questions or concerns about our paper.
> > >
> > > Thank you once again for your time and valuable comments!
> > >
> > > Best Regards,
> > >
> > > Authors

---

### Official Review · Reviewer_7yD4 · 2024-11-03

**Soundness:** 3
**Presentation:** 3
**Contribution:** 2
**Rating:** 6
**Confidence:** 3

**Summary:**

In this paper, the authors studied Bayesian Neural Networks (BNNs) and proposed a new inference method by combining two existing ones: the functional variational inference (fVI) and functional Markov chain Monte Carlo (fMCMC). Their hybrid approach alters between two stages: a V-stage that performs fVI to maximize the ELBO, and a M-stage that performs stochastic gradient fMCMC. They provide theoretical justification that the functional Langevin SDE and the probability flow ODE share the same probability evolution marginals. In the experimental evaluations, the authors show that their approach outperform against baselines and improved predictive accuracy and uncertainty characterization in multiple tasks.

**Strengths:**

The idea from this paper is well-motivated and to my best knowledge, this combination of functional variational inference and functional MCMC is a novel approach. In general, this paper is well-structured, and they provided detailed derivation to show that the functional Langevin stochastic differential equation in the fMCMC and the probability flow ordinary differential equation share the same probability marginals. The math derivation is sound in my opinion.

**Weaknesses:**

I am a bit confused about how to validate the iterations between the V-stage and the M-stage. As the paper mentioned in the page 4, there is a problem when altering between these two stages, because the M-stage outputs a non-parametric distribution while the V-stage requires a parametric one as its input. I wonder how the transformation from $Q_{fs1}$ to $Q_{fs2}$ is defined specifically.

**Questions:**

In the section 5.1, the authors said they make fair comparison between their iterative method and the non-iterative methods by specifying certain number of epochs and iterations. I wonder how those number are determined in terms of the same computational budget between all methods?

---

> ### Author Response · Authors · 2024-11-24
>
> We express our gratitude to Reviewer 7yD4 for positive and insightful feedback. In response to your comments, we prepared a revised version of the manuscript and address specific questions below.
>
> Questions:
>
> $\textbf{Q: }$ In the section 5.1, the authors said they make fair comparison between their iterative method and the non-iterative methods by specifying certain number of epochs and iterations. I wonder how those number are determined in terms of the same computational budget between all methods?
>
> $\textbf{Reply: }$ Our experimental tasks align with the benchmark tasks commonly used in the literature on functional and parameter posterior inference for BNNs. Initially, we referred to the general experimental settings reported in prior studies to define a broad range of training iterations. Through pre-training or initial trials, we selected specific configurations that enable all baseline methods to achieve satisfactory results (e.g., in the extrapolation example in sec.5.1, all methods appeared to converge with more than 10,000 training epochs). This study's experiments are designed to show the different performances between different parameter/function-space inference methods for BNNs. Hence, we do not thoroughly adjust such factors to obtain the best performance, and we just set the same number for all methods for fair comparison. To analyse computational complexity, we provide running time comparisons for the same 2000 training iterations of all inference methods in the uci regressions in Tab4 of Appendix G.2, where our method demonstrates high convergence speed and training stability.
>
> Weaknesses:
>
> $\textbf{W: }$ I am a bit confused about how to validate the iterations between the V-stage and the M-stage. As the paper mentioned in the page 4, there is a problem when altering between these two stages, because the M-stage outputs a non-parametric distribution while the V-stage requires a parametric one as its input. I wonder how the transformation from $Q_{fs1}$ to $Q_{fs2}$ is defined specifically.
>
> $\textbf{Reply: }$ Based on the $Q_{fs1}$ with corresponding explicit posterior over parameters as $q_{s1}(w_{s1}; \theta_{s1})$ obtained in the last V-Stage, we can fit an approximated parametric transformation in T-Stage for the posterior $Q_{fs2}$ obtained in the M-Stage with implicit parameter posterior $q_{s2}(w_{s2})$. Specifically, 1) we first collect function samples $f_{s2_{{1:N}}}^{X_M} \sim Q_{fs2}$ based on the corresponding parameter samples $w_{s2_{{1:N}}}$ obtained from M-Stage, where $X_{M}$ is the randomly sampled measurement set; 2) Based on the local linearization (Rudner et al., 2022), approximating functional $Q_{f_{s1}}$ as a Gaussian process given by  $GP(f_{s1}|f_{s1}(\cdot, \mu_{s1}), J_{\mu_{s1}}\Sigma_{s1}J_{\mu_{s1}}^{T})$, where $J_{\mu_{s1}}$ denotes the Jacobian $\frac{\partial f_{s1}(\cdot, w_{s1})}{\partial w_{s1}}| {w_{s1} = \mu_{s1}}$, and the finite marginal distribution of $Q_{f_{s1}}$ is reduced to an analytical Gaussian distribution; 3) with the three elaborately designed invertible bijections in eq(7)-(9), where the arbitrarily flexible bias function $b(x, w)$ is parametrized as a $2 \times 100$ fully connected nn to achieve high expressive ability, maximizing the log-likelihood in eq(5) to optimize the bijection parameters $\lambda = [a, w]$ (we implement an automatic selection mechanism that chooses one of the three bijections with the highest likelihood during the optimization). Finally, we can obtain the transformed $F_{\lambda_{1}} \circ Q_{f_{s1}}$ for $Q_{fs2}$ with the optimal bijective function $F_{\lambda_{1}}$.

---

> > ### Comment · Reviewer_7yD4 · 2024-11-27
> >
> > Thanks for the detailed reply, particularly the explanation on the V-stage and M-stage. I am convinced that this is a mathematically valid algorithm now. I will raise my score to 6.

---

> ### Author Response · Authors · 2024-11-29
> **Thank you!**
>
> Dear Reviewer 7yD4,
>
> Thank you very much for taking the time to review our responses. We are glad to hear that we have effectively addressed your concerns. I sincerely thank you for your thoughtful feedback and for assigning a higher score to our work.
>
> Thank you once again for your support!
>
> Best Regards,
>
> Authors

---

### Official Review · Reviewer_K6pQ · 2024-11-04

**Soundness:** 4
**Presentation:** 3
**Contribution:** 3
**Rating:** 8
**Confidence:** 4

**Summary:**

This paper proposes a Bayesian inference method for determining the posterior distribution of neural networks that assume a prior distribution over functions rather than parameters. The specific method introduced is a hybrid between functional variational inference and functional Markov Chain Monte Carlo. This is done by alternating between maximizing the functional ELBO in conjunction with a parametric approximate posterior and taking MCMC samples. The gap between these two paradigms are smoothed over by approximating the empirical distribution of posterior samples with a learned, parametric transformation of the variational distribution.

**Strengths:**

The proposed method does a good job of ameliorating the downsides to both fVI and fMCMC while seemingly retaining the benefits of both. I found the learning of a parametric representation of the empirical distribution (in the T-Stage) with a normalizing flow to be particularly clever at bridging the gap between VI and MCMC. The proposed approach makes a solid contribution towards Bayesian methods for neural networks, as functional priors in general are an attractive approach in this space but previously have been plagued with practical downsides that this technique improves upon.

**Weaknesses:**

I mainly found parts of the experimental results to be a bit lacking. In the abstract, the paper claims that the proposed method improves upon the restrictive posterior of fVI and the slow mixing times of fMCMC; however, as far as I can tell only the former has been showcased. A more expressive posterior can shown with visualizations (Fig 1) or could be argued with improved predictive performance (rest of Section 5), whereas the mixing times, especially in high dimensions, requires a stand alone investigation. Having another synthetic experiment where the posterior distribution is known (e.g., Gaussian Process) and comparing how many iterations is required to converge to arbitrary precision would demonstrate this well. Additionally, showcasing some loss-curves and other metrics throughout training for the real-world experiments could also showcase this.

**Questions:**

I do not believe this general approach has been taken for parametric Bayesian neural networks, is this the case? If not, do you see any potential for also adapting this scheme (V->M->T->...) for parameter-based inference?

---

> ### Author Response · Authors · 2024-11-24
>
> We express our gratitude to Reviewer K6pQ for positive and insightful feedback. In response to your comments, we prepared a revised version of the manuscript and address specific questions below.
>
> Questions:
>
> $\textbf{Q: }$ I do not believe this general approach has been taken for parametric Bayesian neural networks, is this the case?
>
> $\textbf{Reply: }$ Our method can indeed be applied to parameter space inference in a similarly smooth manner, specifically by simply replacing the fVI optimization in the V-Stage and the fMCMC procedure in the M-Stage with the corresponding parameter-space VI and MCMC, respectively, which reflects the universality of our proposed hybrid algorithm.
>
> Weaknesses:
>
> $\textbf{W: }$ The mixing times of fMCMC, especially in high dimensions, require a stand-alone investigation.  Additionally, showcasing some loss curves and other metrics throughout training for real-world experiments could also showcase this.
>
> $\textbf{Reply: }$ Thanks for the suggestion! We add the analysis of mixing Time in Appendix G.1. We plot the trajectories of fMCMC in the M-Stage, the parameter-space SGLD and the functional fSGLD for the synthetic extrapolation and uci regression (Yacht) in Fig6. The parameter dimension of the nn used for the extrapolation experiments is 10401 dimensions, and the parameters of the uci regression model are 191 dimensions. With the aid of the explicit optimization of fVI in the V-stage, which quickly converges to local high-probability target regions, the fMCMC in the M-stage of our hybrid FVIMC exhibits a rapid mixing rate for high-dimensional parameter space. In the extrapolation example, the fMCMC in the M-stage converges rapidly to the stationary measure within 800 iterations, while fSGLD and SGLD show stabilizing trends until after 2000 iterations. Similarly, our FVIMC reaches the stationary for the uci regression in only 400 iterations, while SGLD and fSGLD require more than 500 iterations for mixing. Additionally, the convergence processes of training loss for all VI baselines and the fVI in the V-Stage of FVIMC are shown in Fig7 in Appendix G.2, where FVIMC achieves high convergence speed and training stability.

---

> > ### Comment · Reviewer_K6pQ · 2024-11-29
> > **Reply**
> >
> > Thank you for this additional work and clarification. After reading the the initial response to my questions, as well as the other reviews and replies in general, I will raise my score to an 8 as I believe this to be an interesting and novel work. I enjoyed the paper and hope to see it polished some more and published soon.

---

> > > ### Author Response · Authors · 2024-12-01
> > > **Thank you!**
> > >
> > > Dear Reviewer K6pQ,
> > >
> > > Thank you very much for taking the time and effort to review our manuscript and responses. We are glad to hear that we have effectively addressed your concerns. Your thoughtful feedback has greatly contributed to improving the quality of our work.
> > >
> > > Once again, we deeply appreciate your improved score and support!
> > >
> > > Best Regards,
> > >
> > > Authors

---

### Official Review · Reviewer_bU3Q · 2024-11-07

**Soundness:** 3
**Presentation:** 2
**Contribution:** 2
**Rating:** 5
**Confidence:** 3

**Summary:**

This paper proposes a method for functional posterior inference in BNNs, combining the best of functional variational inference and functional MCMC. It alternates the two, with an additional step of fitting a parametric transformation to the MCMC samples to form the analytic approximate posterior for the next round. Experiments show that the method achieve better performance than other BNN techniques.

**Strengths:**

The method seems technically sound.

The comparisons to other BNN methods are reasonably broad ranging and convincing.

**Weaknesses:**

The method and general area are technically very involved. While I have not checked everything, things do make sense at high level though there are some notational inconsistencies.

The write-up is very involved as a result. I think the method can probably be applied in the more simple non-functional setting, which can make the description more easy to understand and get across.

Experiments show that the method achieve better performance than other BNN techniques. However the paper did not compare against other baselines that are "not as Bayesian", though these are often simpler, computationally more efficient, and more performant empirically. These include deep ensembles, SNGPs, DUE, epinets, temperature scaling, or even conformable prediction methods.

It would also be nice to see experimental results on uncertainty quantification, e.g. reporting expected calibration error for classification, given as UQ is one of the selling points of Bayesian methods.

My worry with the paper is the technical complexity and computationally expense, that while shown to be better than simpler BNN baselines, has not been shown to be better than non BNN methods on a representative benchmark (e.g. uncertainty quantification). This calls into question the practical impact and usefulness of the method.  Granted this criticism can perhaps be directed at many papers in this area, but I think as researchers it is good for us to be able to answer the question of why/under what circumstance somebody should use the methods/tools that we have developed.

**Questions:**

1. Which specific fVI method do you use? I didn't find any details of this in the paper, while the algorithm in appendix is quite inpenetrable with some notation undefined etc. This seems quite an important question, as I don't know how the algorithm works with the complex posterior forms composed of the transformations with Q_fs1.

2. I think Q_fsK in Prop 3.1 should be Q_fs1?

3. The forms of the invertible transformations are very simple. Can they really capture sufficiently complex posteriors? It would be nice to demonstrate this in a simple 2D toy example?

4. I don't think there's a precise statement of Prop 3.2. Can you furnish this? I think it's saying that if you look at gradient flow of the ELBO, you also get the evolution of the marginal distributions under Langevin SDE. I don't think it really says that for a particular (implementable) fVI method, that they are the same, right? I feel that Prop 3.2 is just saying that the fVI and fMCMC are "going in the same direction".

5. The transformation step will probably introduce errors, and if we want to careful it is important to quantify these errors, and whether they are controlled or will compound as the algorithm runs.

---

> ### Author Response · Authors · 2024-11-24
>
> We express our gratitude to Reviewer bU3Q for the insightful feedback. In response to your comments, we prepared a revised version of the manuscript and address specific questions below.
>
> $\textbf{Q1: }$  Which specific fVI method do you use?
>
> $\textbf{Reply: }$ Eq.(1) in the main paper is the general KL-based functional ELBO to be maximized in fVI. However, considering the potential definitional and estimation problems of kL divergence for distribution over functions, we use the alternative Wasserstein distance-based variational objective with an  additional enhanced second-order moment
> matching term for fVI [1] in practical optimization as:
> $L_{W} = -\frac{1}{M} \sum_{(x, y) \in \mathcal{B}}E_{Q_{f}}[\log p(y \mid f(x;\mathbf{w}))] +  W_1(Q_{f} (f^{X_{M}}), P_{0} (f^{X_{M}})) + |var(Q_{f}(f^{X_{M}}))-var(P_0(f^{X_{M}}))|$. It is a fully sampling-based procedure that does not depend on the closed form of $Q_{f}$ and is more flexible than the KL-based objective. We will carefully check the notation in the pseudocode in the final version.
>
> [1] Wu et al. Functional Wasserstein bridge inference for Bayesian deep learning. UAI, 2024.
>
> $\textbf{Q2: }$ I think $Q_{fsk}$ in Prop 3.1 should be $Q_{fs1}$?
>
> $\textbf{Reply: }$ Actually, it should be $Q_{fsK}$ since $Q_{fsi}$ is continuously updated. Based on the transformed posterior $F_{\lambda_{1}} \circ Q_{f_{s1}}$ obtained in the first T-Stage1 of the main paper for the posterior  $Q_{fs2}$ of samples from the second M-Stage2, we can continue to perform the next V-Stage3, and the resulting approximate posterior measure is denoted as $F_{\lambda_{1}} \circ Q_{f_{s3}}$ with the corresponding updated posterior over parameters as $q_{s3}(w_{s3}; \theta_{s3})$. In the following M-Stage4, the probability measure of the collected function samples is denoted as $F_{\lambda_{1}} \circ Q_{f_{s4}}$ and the corresponding implicit posterior over parameters is $q_{s4}(w_{s4})$. We perform the second T-sage2 to transform this $F_{\lambda_{1}} \circ Q_{f_{s4}}$ into $F_{\lambda_{2}} \circ F_{\lambda_{1}} \circ Q_{f_{s3}}$ based on the $F_{\lambda_{1}} \circ Q_{f_{s3}}$ with parametric $q_{s3}(w_{s3}; \theta_{s3})$ obtained in the last V-Stage3, then it is able to run the next V-Stage5 smoothly and obtain the updated $F_{\lambda_{2}} \circ F_{\lambda_{1}} \circ Q_{f_{s5}}$ with the corresponding $q_{s5}(w_{s5}; \theta_{s5})$. We continue in this fashion by successively alternating between V-Stage and M-Stage with intermediate T-Stage, so the final approximate posterior measure is in a composite form: $F_{\lambda_{(K-1)/2}} \circ \cdots F_{\lambda_{2}} \circ F_{\lambda_{1}} \circ Q_{f_{sK}}$ with a total $K$ V-Stages and M-Stages. The base parametric $q_{si}(w_{si}; \theta_{si})$ is being continuously updated, so the corresponding final base posterior over functions is no longer $Q_{fs1}$, but should be $Q_{fsK}$. You can see Appendix B.1 for a more detailed derivation of Prop 3.1.
>
> $\textbf{Q3: }$ The forms of the invertible transformations are very simple. Can they really capture sufficiently complex posteriors?
>
> $\textbf{Reply: }$ The bias function $b(x; \omega)$ in the invertible transformations can be arbitrarily complex since the computation of Jacobian does not depend on it. So, we can form $b(x; \omega)$ as neural networks parametrized by $\omega$ with
> arbitrarily flexible structures to fit the complex posteriors. For example, in our experiment settings, we made it a two-hidden-layer fully connected neural network, and each layer with 100 hidden units, which is flexible and expressive to fit the complex posteriors.
>
> $\textbf{Q4: }$ I don't think there's a precise statement of Prop 3.2. Can you furnish this? I feel that Prop 3.2 is just saying that the fVI and fMCMC are "going in the same direction".
>
> $\textbf{Reply: }$ We extend the parameter-space conclusion that Langevin SDE and the probability flow ODE derived from the Wasserstein gradient of VI share the same marginals $\{q_{t}(\mathbf{w})\}_{t\geq0}$ [2][3] to function space. Similar to the parameter space derivation process, we derived that the Wasserstein gradient flow of fVI reads the Fokker-Planck equation of functional Langevin SDE, and further proved that they share the same probability measure evolution marginals.
>
> [2] Hoffman et al. Black-box variational inference as a parametric approximation to Langevin dynamics. ICML, 2020.
>
> [3] Yi et al. Bridging the gap between variational inference and Wasserstein gradient flows. arXiv:2310.20090, 2023.

---

> ### Author Response · Authors · 2024-11-24
>
> $\textbf{Q5: }$ The transformation step will probably introduce errors.
>
> $\textbf{Reply: }$ Thanks for the suggestion! We add the plot of the rmse comparisons between the posterior in the M-Stage and the transformed posterior obtained in the T-Stage in the uci regressions to quantify the transformation errors in Appendix G.4. We can see that as the algorithm proceeds, this transformation error can be significantly reduced and effectively controlled. Additionally, the training losses in the T-stages converge quickly and smoothly. Therefore, the parameterized posterior obtained by the T-Satge transformation is a reasonable approximation to the implicit posterior of M-Stage.
>
> Weaknesses:
>
> $\textbf{W1: }$ I think the method can probably be applied in the more simple non-functional setting, which can make the description more easy to understand and get across.
>
> $\textbf{Reply: }$ Our approach can indeed be applied to parameter-space inference smoothly, but this paper is concerned with the functional posterior inference for BNNs and is dedicated to combining fMCMC and fVI to inherit their respective strengths effectively. We will carry out careful proofreading work and further improve the expression of some sentences in the final version.
>
> $\textbf{W2: }$ It would also be nice to see experimental results on uncertainty quantification.
>
> $\textbf{Reply: }$ We add the plot of calibration curves of all parameter/function-space inference approaches for the synthetic extrapolation example in Appendix G.3, which assesses how well the predicted probabilities of a model match the true frequencies. Our FVIMC demonstrates competitive calibration capability. We think the strong and reasonable uncertainty presented in Fig1(a), and the favourable nll and auc results in Tab2 and Tab3 in the main paper can also indicate the reliable uncertainty quantification of our method.
>
> $\textbf{W3: }$ The paper did not compare against other baselines that are "not as Bayesian".
>
> $\textbf{Reply: }$ In this study, our experiments are mainly designed to demonstrate the advantages of our hybrid method compared with other parameter/function-space VI and MCMC baselines. Our baselines are consistent with those used in the existing literature on functional /parameter posterior inference for BNNs. We will consider extending the comparisons with other non-Bayesian methods in our following study.
>
> $\textbf{W4: }$ My worry with the paper is the technical complexity and computational expense.
>
> $\textbf{Reply: }$ To compare the efficiency between FVIMC and other inference approaches, we provide running time comparisons for all 2000 training epochs of all inference methods in the uci regression on the small Boston dataset(455 training points) and the large Protein dataset(41157 training points) in Tab4 in Appendix G.2. In the small Boston dataset, our FVIMC shows similar computational efficiency to the other functional fVI, GWI, IFBNN, and is nearly 5 times faster than FBNN. With the fast convergence of the V-Stage to locally high probabilities, our FVIMC demonstrates its computational efficiency advantage on the larger Protein dataset: the running time of GWI and FBNN is nearly 1.7-2.5× higher than FVIMC. The mixing rate of fSGLD, which performs well on small datasets, has deteriorated at this time and is significantly slower than our hybrid method. Moreover, the convergence processes of training loss for all VI baselines and the V-Stage of FVIMC are shown in Fig7, where FVIMC achieves high convergence speed and training stability.

---

> ### Author Response · Authors · 2024-11-29
>
> Dear Reviewer bU3Q,
>
> Thank you very much for taking the time and effort to review our manuscript, which has helped us refine our work. As the discussion phase between authors and reviewers is nearing its conclusion, we sincerely hope that the revisions and responses have clarified your concerns. If you have any remaining doubts or require further clarification, please do not hesitate to let us know. We are more than willing to continue the discussion to address them.
>
> Thank you once again for your time and valuable suggestions.
>
> Best Regards,
>
> Authors

---

### Meta-Review · Area_Chair_QY35 · 2024-12-21

**Metareview:**

This paper proposes a hybrid method for functional posterior inference in BNNs, combining functional VI and functional MCMC through an alternating scheme, achieving improved performance over BNN baselines. The reviews found the proposed hybrid method novel, sound, and a solid contribution to BNN research with empirical support across various tasks. The reviewers had concerns about the presentation being very complex, lack of comparison to non-BNN baselines, further empirical evaluation in terms of uncertainty quantification, limited exploration of hyperparameter tuning, and concerns about the applicability and validity of Proposition 3.2 under certain conditions, including data subsampling and Wasserstein distance replacement. The authors addressed most reviewer concerns, providing clarifications on the theoretical aspects and additional empirical results. Given the theoretical advancements and empirical performance and the fact that the majority of the reviews recommend accepting the paper, I recommend accepting the paper.

**Additional Comments On Reviewer Discussion:**

As described in the meta-review, the authors addressed most of the reviewers' concerns during the rebuttal and discussion, providing clarifications on the theoretical aspects and additional empirical results.

---

### Decision · Program_Chairs · 2025-01-22

Accept (Poster)